# Mushroom body output neurons encode valence and guide memory-based action selection in *Drosophila*

Yoshinori Aso[1]*, Divya Sitaraman[1,2,6,7]†, Toshiharu Ichinose[3,4], Karla R Kaun[1]‡, Katrin Vogt[3], Ghislain Belliart-Guérin[5], Pierre-Yves Plaçais[5], Alice A Robie[1], Nobuhiro Yamagata[3,4], Christopher Schnaitmann[3§], William J Rowell[1], Rebecca M Johnston[1], Teri-T B Ngo[1], Nan Chen[1], Wyatt Korff[1], Michael N Nitabach[1,2,6,7], Ulrike Heberlein[1], Thomas Preat[5], Kristin M Branson[1], Hiromu Tanimoto[3,4], Gerald M Rubin[1]*

[1]Janelia Research Campus, Howard Hughes Medical Institute, Ashburn, United States; [2]Department of Cellular and Molecular Physiology, Yale School of Medicine, New Haven, United States; [3]Max Planck Institute of Neurobiology, Martinsried, Germany; [4]Graduate School of Life Sciences, Tohoku University, Sendai, Japan; [5]Genes and Dynamics of Memory Systems, Brain Plasticity Unit, Centre National de la Recherche Scientifique, ESPCI, Paris, France; [6]Department of Genetics, Yale School of Medicine, New Haven, United States; [7]Program in Cellular Neuroscience, Neurodegeneration and Repair, Yale School of Medicine, New Haven, United States

*For correspondence: asoy@ janelia.hhmi.org (YA); rubing@ janelia.hhmi.org (GMR)

Present address: †Department of Psychological Sciences, University of San Diego, San Diego, United States; ‡Department of Neuroscience, Brown University, Providence, United States; §Department of Neurobiologie and Tierphysiologie, Institute of Biology 1, Albert Ludwig University of Freiburg, Freiburg, Germany

Competing interests: The authors declare that no competing interests exist.

**Abstract** Animals discriminate stimuli, learn their predictive value and use this knowledge to modify their behavior. In *Drosophila*, the mushroom body (MB) plays a key role in these processes. Sensory stimuli are sparsely represented by ~2000 Kenyon cells, which converge onto 34 output neurons (MBONs) of 21 types. We studied the role of MBONs in several associative learning tasks and in sleep regulation, revealing the extent to which information flow is segregated into distinct channels and suggesting possible roles for the multi-layered MBON network. We also show that optogenetic activation of MBONs can, depending on cell type, induce repulsion or attraction in flies. The behavioral effects of MBON perturbation are combinatorial, suggesting that the MBON ensemble collectively represents valence. We propose that local, stimulus-specific dopaminergic modulation selectively alters the balance within the MBON network for those stimuli. Our results suggest that valence encoded by the MBON ensemble biases memory-based action selection.

## Introduction

To survive in a dynamic environment, an animal must discover and remember the outcomes associated with the stimuli it encounters. It then needs to choose adaptive behaviors, such as approaching cues that predict food and avoiding cues that predict danger. The neural computations involved in using such memory-based valuation of sensory cues to guide action selection require at least three processes: (1) sensory processing to represent the identity of environmental stimuli and distinguish among them; (2) an adaptive mechanism to assign valence—positive or negative survival value—to a sensory stimulus, store that information, and recall it when that same stimulus is encountered again; and (3) decision mechanisms that receive and integrate information about the valence of learned stimuli and then bias behavioral output. To understand such decision-making processes, one approach is to locate the sites of synaptic plasticity underlying memory formation, identify the postsynaptic neurons that transmit stored information to the downstream circuit and discover how their altered activities bias behavior.

**eLife digest** An animal's survival depends on its ability to respond appropriately to its environment, approaching stimuli that signal rewards and avoiding any that warn of potential threats. In fruit flies, this behavior requires activity in a region of the brain called the mushroom body, which processes sensory information and uses that information to influence responses to stimuli.

Aso et al. recently mapped the mushroom body of the fruit fly in its entirety. This work showed, among other things, that the mushroom body contained 21 different types of output neurons. Building on this work, Aso et al. have started to work out how this circuitry enables flies to learn to associate a stimulus, such as an odor, with an outcome, such as the presence of food.

Two complementary techniques—the use of molecular genetics to block neuronal activity, and the use of light to activate neurons (a technique called optogenetics)—were employed to study the roles performed by the output neurons in the mushroom body. Results revealed that distinct groups of output cells must be activated for flies to avoid—as opposed to approach—odors. Moreover, the same output neurons are used to avoid both odors and colors that have been associated with punishment. Together, these results indicate that the output cells do not encode the identity of stimuli: rather, they signal whether a stimulus should be approached or avoided. The output cells also regulate the amount of sleep taken by the fly, which is consistent with the mushroom body having a broader role in regulating the fly's internal state.

The results of these experiments—combined with new knowledge about the detailed structure of the mushroom body—lay the foundations for new studies that explore associative learning at the level of individual circuits and their component cells. Given that the organization of the mushroom body has much in common with that of the mammalian brain, these studies should provide insights into the fundamental principles that underpin learning and memory in other species, including humans.

The mushroom body (MB) is the main center of associative memory in insect brains (*de Belle and Heisenberg, 1994*, *Heisenberg et al., 1985*; *Dubnau et al., 2001*; *McGuire et al., 2001*). While the MB processes several modalities of sensory information and regulates locomotion and sleep (*Martin et al., 1998*; *Liu et al., 1999*; *Joiner et al., 2006*; *Pitman et al., 2006*; *Zhang et al., 2007*; *Hong et al., 2008*; *Vogt et al., 2014*), MB function has been most extensively studied in the context of olfactory memory—specifically, associating olfactory stimuli with environmental conditions in order to guide behavior. In *Drosophila*, olfactory information is delivered to the MB by projection neurons from each of ~50 antennal lobe glomeruli (*Marin et al., 2002*; *Wong et al., 2002*; *Jefferis et al., 2007*; *Lin et al., 2007*; *Vosshall and Stocker, 2007*; *Yu et al., 2010*). Connections between the projection neurons and the ~2000 Kenyon cells (KCs), the neurons whose parallel axonal fibers form the MB lobes (*Crittenden et al., 1998*; *Aso et al., 2009*), are not stereotyped (*Figure 1A*) (*Murthy et al., 2008*; *Caron et al., 2013*); that is, individual flies show distinct wiring patterns between projection neurons and KCs. Sparse activity of the KCs represents the identity of odors (*Laurent and Naraghi, 1994*; *Perez-Orive et al., 2002*; *Turner et al., 2008*). The output of the MB is conveyed to the rest of the brain by a remarkably small number of neurons—34 cells of 21 cell types per brain hemisphere (*Figure 1B*, *Table 1*) (*Aso et al., 2014*).

The information flow from the KCs to the MB output neurons (MBONs) has been proposed to transform the representation of odor identity to more abstract information, such as the valence of an odor based on prior experience (See discussion in *Aso et al., 2014*). In contrast to KCs, MBONs have broadly tuned odor responses; any given odor results in a response in most MBONs, although the magnitude of the response varies among MBON cell types (Hige et al., unpublished) (*Cassenaer and Laurent, 2012*). Unlike the stereotyped response to odors of the olfactory projection neurons that deliver odor information to the MB, the odor tuning of the MBONs is modified by plasticity and varies significantly between individual flies, suggesting that MBONs change their response to odors based on experience (Hige et al., unpublished).

For olfactory associative memory in *Drosophila*, multiple lines of evidence are consistent with a model in which dopamine-dependent plasticity in the presynaptic terminals of KCs alters the strength of synapses onto MBON dendrites. This is thought to provide a mechanism by which the response of MBONs to a specific odor could represent that odor's predictive value. D1-like dopamine receptors and components of the cAMP signaling pathway, such as the $Ca^{2+}$/Calmodulin-responsive adenylate

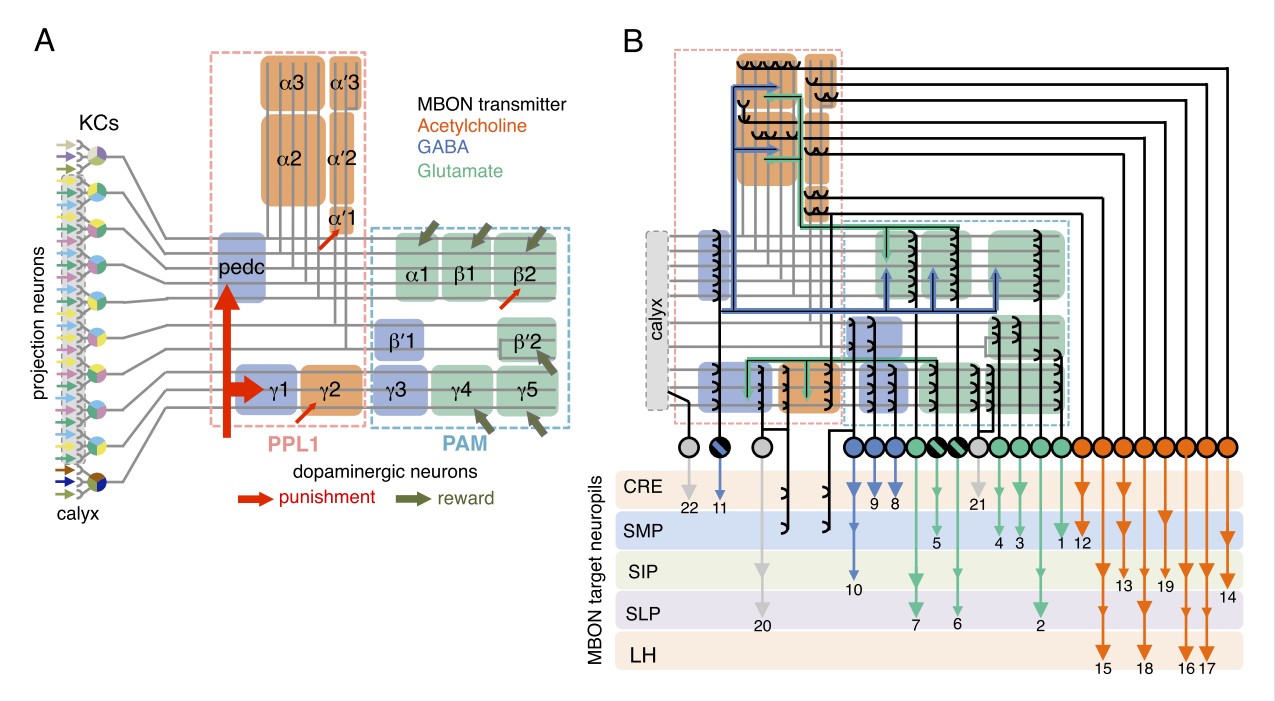

**Figure 1**. Circuit diagrams of the mushroom body. (**A**) The innervation patterns of extrinsic neurons define 15 compartments in the MB lobes and one compartment in the core of distal pedunculus (pedc); the compartments are represented by rectangles that are color-coded based on the neurotransmitter used by the mushroom body output neurons (MBONs) having dendrites in that compartment (green, glutamate; blue, GABA; orange, acetylcholine). Projection neurons (far left, colored arrows) from the antennal lobe convey olfactory sensory information to the MB calyx where they synapse on the dendrites of Kenyon cells (KCs). The parallel axon fibers of the KCs (gray lines) form the lobes (α/β, α'/β' and γ) where KCs terminate onto the dendrites of the MBONs. Each of the seven types of KCs innervates a specific layer within a given lobe. The dendrites of individual MBON types and the terminals of dopaminergic neurons (DANs) intersect the longitudinal axis of KC axon-bundles in specific compartments along the lobes. MBONs using the same transmitter are spatially co-localized in the lobes (See *Figure 1B*). Innervation areas of PPL1 and PAM cluster DANs axons in the MB lobes are indicated by the rectangles outlined in dashed lines. Activation of subsets of DANs can convey punishment or reward, respectively inducing aversive or appetitive memory when activation is paired with odor presentation. The size of arrows indicates magnitude of memory induced by DAN activation. See text for references. (**B**) Schematic representation of the 21 cell types of MBONs in the lobes and one cell type of MBON in the calyx based on the data presented in the accompanying manuscript (*Aso et al., 2014*): circles, cell bodies; semicircles, dendrites; arrowheads, axon terminals; color-coding is by neurotransmitter as in panel (**A**) Three MBON cell-types (GABAergic MBON-γ1pedc>α/β, glutamatergic MBON-γ4>γ1γ2 and MBON-β1>α; marked as 11, 5 and 6 respectively) send axons into the MB lobes. Axons of MBON-γ4>γ1γ2 project from γ4 to γ1 and γ2, and thus have the potential to affect activity of MBON-γ1pedc>α/β. From γ1, the axon of MBON-γ1pedc>α/β projects to compartments in the α/β lobes including β1, where dendrites of MBON-β1>α arborize. Axons of both MBON-γ1pedc>α/β and MBON-β1>α project to the compartments in the α lobe. Therefore activity of MBONs in the α lobe can be regulated by these layered inter-compartmental connections. These three types of MBONs (11, 5 and 6) do not project back to their own dendrites. Therefore, the organization of the MBONs can be viewed as forming a multilayered feed-forward network (*Aso et al., 2014*). MBONs project to a small number of brain areas: the crepine (CRE; a region surrounding the horizontal/medial lobes), the superior medial protocerebrum (SMP), superior intermediate protocerebrum (SIP) and superior lateral protocerebrum (SLP) and the lateral horn (LH). The size of the arrowhead reflects the relative number of termini in each area. The MBONs are numbered and listed in *Table 1*. See the accompanying manuscript (*Aso et al., 2014*) and *Table 1* for details.

cyclase encoded by the *rutabaga* gene, are required specifically in the KCs for memory formation (*Livingstone et al., 1984*; *Zars et al., 2000*; *Schwaerzel et al., 2003*; *Kim et al., 2007*; *McGuire et al., 2003*; *Gervasi et al., 2010*; *Qin et al., 2012*) and *rutabaga* was shown to be required for the establishment of the differences in MBON odor tuning between individuals (Hige et al., unpublished). Reward and punishment recruit distinct sets of dopaminergic neurons (DANs) that project to specific regions in the MB lobes (*Mao and Davis, 2009*; *Burke et al., 2012*; *Liu et al., 2012*). Moreover, exogenous activation of these DANs can substitute for reinforcing stimuli to induce either appetitive or aversive memory, depending on DAN cell type (*Figure 1A*) (*Yamagata et al., in press*, *Perisse et al., 2013*; *Schroll et al., 2006*; *Claridge-Chang et al., 2009*; *Aso et al., 2010, 2012*; *Liu et al., 2012*; *Burke et al., 2012*). In sum, while the identity of the learned odor is likely encoded by the small subset of KCs activated by that odor, whether dopamine-mediated modulation assigns positive or negative

**Table 1.** The list of MBON cell types

| cell type name | cell body cluster | putative transmitter | Number of cells | short names (number in *Figure 1B*) | names in previous publications |
|---|---|---|---|---|---|
| MBON-γ5β'2a | M4/M6 cluster | glutamate | 1 | MBON-01 | MB-M6 |
| MBON-β2β'2a | | | 1 | MBON-02 | |
| MBON-β'2mp | | | 1 | MBON-03 | MB-M4 |
| MBON-β'2mp_bilateral | | | 1 | MBON-04 | MB-M4? |
| MBON-γ4>γ1γ2 | MV2 cluster | | 1 | MBON-05 | |
| MBON-β1>α | | | 1 | MBON-06 | MB-MV2 |
| MBON-α1 | | | 2 | MBON-07 | |
| MBON-γ3 | γ3 cluster | GABA | 1 | MBON-08 | |
| MBON-γ3β'1 | | | 1 | MBON-09 | |
| MBON-β'1 | | | 8 | MBON-10 | |
| MBON-γ1pedc>α/β | | | 1 | MBON-11 | MB-MVP2 |
| MBON-γ2α'1 | V3/V4 cluster | acetycholine | 2 | MBON-12 | |
| MBON-α'2 | | | 1 | MBON-13 | MB-V4 |
| MBON-α3 | | | 2 | MBON-14 | MB-V3 |
| MBON-α'1 | V2 cluster | | 2 | MBON-15 | |
| MBON-α'3ap | | | 1 | MBON-16 | MB-V2α' |
| MBON-α'3m | | | 2 | MBON-17 | MB-V2α' |
| MBON-α2sc | | | 1 | MBON-18 | MB-V2α |
| MBON-α2p3p | | | 2 | MBON-19 | |
| MBON-γ1γ2 | | N.D. | 1 | MBON-20 | |
| MBON-γ4γ5 | | | 1 | MBON-21 | |
| MBON-calyx | | | 1 | MBON-22 | MB-CP1 |

The name of each cell type is given. For MBONs, we use MBON plus the name of MB lobe compartment(s) in which their dendrites arborize (for example, MBON-α1 for the MB output neurons that have dendrites in the α1 compartment; MBON-γ5β'2a for the MB output neurons that have dendrites in the γ5 compartment and the anterior layer of the β'2 compartment). For the three classes of MBONs which also have *axon* terminals in the MB lobes, the '>' symbol is followed by the compartments (or lobes) in which the axons terminate (for example, MBON-β1>α for the MB output neuron with dendrites in β1 and synaptic terminals in the α lobe.). Subsets of MBONs have their cell bodies clustered, presumably reflecting their common developmental origin; these clusters are indicated. The neurotransmitter used by each MBON (where known) is indicated as is the number of cells of that type found per brain hemisphere (data from the accompanying manuscript; (***Aso et al., 2014***). We also provide a short cell type name based on simple numbering; the number of an MBON corresponds to number shown in *Figure 1B*. Many of these cell types have been previous described under alternative names, as indicated (***Tanaka et al., 2008***).

valence to that odor would be determined by where in the MB lobes KC-MBON synapses are modulated and thus which MBON cell types alter their response to the learned odor.

Combining the above observations with our comprehensive anatomical characterization of MB inputs and outputs (***Aso et al., 2014***) lays the groundwork for testing models of how the MB functions as a whole. We suggest that each of the 15 MB compartments—regions along the MB lobes defined by the arborization patterns of MBONs and DANs (see *Figure 1*)—functions as an elemental valuation system that receives reward or punishment signals and translates the pattern of KC activity to a MBON output that serves to bias behavior by altering either attraction or aversion. This view implies that multiple independent valuation modules for positive or negative experiences coexist in the MB lobes, raising the question of how the outputs across all the modules are integrated to result in a coherent, adaptive biasing of behavior.

Although several MBON cell types have been shown to play a role in associative odor memory (*Sejourne et al., 2011*; *Pai et al., 2013*; *Placais et al., 2013*), the functions of most MBONs have not been studied. Based on our anatomical analyses (*Aso et al., 2014*), we believe that just 34 MBONs of 21 types provide the sole output pathways from the MB lobes. To gain mechanistic insight into how the ensemble of MBONs biases behavior, we would first like to know the nature of the information conveyed by individual MBONs and the extent to which their functions are specialized or segregated into different information channels. Then we need to discover how the activities of individual MBONs contribute to influence the behavior exerted by the complete population of MBONs. Thus, in order to understand how memory is translated into changes in behavior, we need to have experimental access to a comprehensive set of MBONs and investigate how the outputs from different MBONs bias behavior, singly and in combination.

In the accompanying paper (*Aso et al., 2014*), we describe the detailed anatomy of the DANs and MBONs (*Figure 1*) and the generation of intersectional split-GAL4 driver lines to facilitate their study. All but one of the 21 MBON cell types consists of only one or two cells per hemisphere (*Table 1*). Dendrites of MBONs that use the same neurotransmitter—GABA, glutamate or acetylcholine—are spatially clustered in the MB lobes. Intriguingly, this spatial clustering resembles the innervation patterns of modulatory input by two clusters of dopaminergic neurons, PPL1 and PAM. MBONs have their axonal terminals in a small number of brain regions, but their projection patterns also suggest pathways for relaying signals between compartments of the MB lobes; three MBONs send direct projections to the MB lobes and several other MBONs appear to target the dendrites of specific DANs.

Our split-GAL4 drivers give us the capability to express genetically encoded effectors in identified MBONs to modify their function. In this study, we examine the roles of specific MBONs in various learning and memory tasks as well as in the regulation of locomotion and sleep. We also studied whether direct activation of specific MBONs are sufficient to elicit approach or avoidance. Our results indicate that the ensemble of MBONs does not directly specify particular motor patterns. Instead, MBONs collectively bias behavior by conveying the valence of learned sensory stimuli, irrespective of the modality of the stimulus or the specific reward or punishment used during conditioning.

## Results

### Strategy for assigning behavioral roles to MBONs

A powerful strategy to discover if a neuronal population plays a role in a particular behavior is to observe the effects of inactivating or activating those neurons. A genetic driver can be used to express an exogenous protein that either promotes or blocks neuronal activity. By repeating such manipulations with a large collection of drivers, each specific for a different set of neurons, one can in principle discover cell types required for a particular behavior. This is analogous to a screen to identify genes that when mutated disrupt a cellular function, but it is the activity of cells—rather than that of genes—being altered. This approach has been widely used in *Drosophila* (Reviewed in *Venken et al., 2011*; *Griffith, 2012*).

There are many challenges in carrying out such an approach. As in all biological systems, we expect extensive resiliency to perturbation. Such robustness might mask the effects of manipulating the activity of a small population of neurons, making them undetectable above the normal variation between animals. In addition to these inherent limitations, the genetic tools at our disposal have often been inadequate. In this study, we have used improved genetic tools and employed several strategies to mitigate their remaining limitations, as detailed below and in 'Materials and methods'. By assaying many different behaviors with the same genetic reagents, we were better able to evaluate the specificity of the behavioral effects we observed. Our experimental design placed an emphasis on avoiding false positives. Extensive analysis was restricted to MBON lines showing a phenotype in the initial screens and our scoring criteria were conservative. Consequently, we are likely to have missed detecting some cell types with effects on the behavior under assay.

### Photoactivation of MBONs can drive either approach or avoidance

We sought to first determine the nature of the information conveyed by MBONs. In the most widely used olfactory conditioning assay, memory is assessed after training by allowing flies to distribute between two arms of a T-maze: one arm perfused with a control odor and the other arm with an odor

that had been previously associated with punishment or reward. If the valence of the learned odor were encoded by the altered activities of specific MBONs, artificial activation of those MBONs in untrained flies in the absence of odor presentation would be expected to result in avoidance or approach behavior that mimicked the conditioned odor response. To test this hypothesis, we used a circular arena in which groups of flies expressing the red-shifted channelrhodopsin CsChrimson (*Klapoetke et al., 2014*) in MBONs were allowed to freely distribute between dark quadrants and quadrants with activating red light (*Figure 2A*; see 'Materials and methods' for details). Activating sensory neurons for $CO_2$ or bitter taste in this manner induced strong avoidance (*Figure 2B*), consistent with previous reports (*Suh et al., 2007*). By testing our collection of MBON split-GAL4 drivers in this assay, we found cell types whose activation resulted in avoidance of the red light and others whose activation led to attraction (*Figure 2C*). Behavioral valence was highly correlated with MBON transmitter type: all MBONs eliciting aversion were glutamatergic and all the MBONs eliciting attraction were either GABAergic or cholinergic. We selected two split-GAL4 drivers for each neurotransmitter type for further analysis, choosing lines that gave robust phenotypes in the initial screening and that showed highly specific expression patterns based on direct assessment of CsChrimson expression (*Figure 3A–F*). The phenotypes of these lines were reproducible (*Figure 3G–I*) and none of the drivers

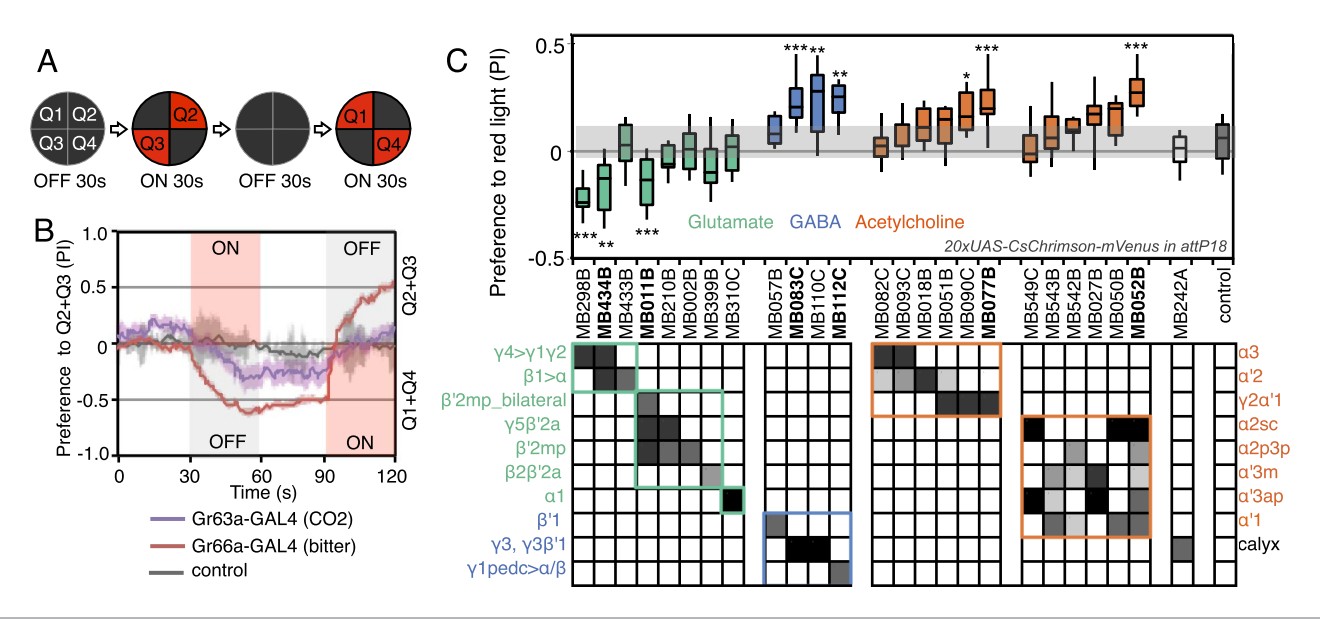

**Figure 2**. Screening MBONs for behavioral valence. (**A**) Behavioral valence assay. Approximately twenty female flies were placed in a 10 cm circular arena in a dark chamber and allowed to distribute themselves freely among the four quadrants. From 30–60 s, two of the quadrants (Q2&3) were illuminated with 617 nm peaked red LED lights to activate CsChrimson-containing neurons; from 90–120 s, the other two quadrants (Q1 & 4) were illuminated instead. No quadrants were illuminated from 0–30 s or from 60–90 s. The flies were video recorded and their locations in the arena were used to calculate behavioral preferences. (**B**) For these time-domain plots, the quadrant preference at each point in time was calculated as [(number of flies in Q2&3) - (number of flies in Q1&4)]/(total number of flies). Flies expressing CsChrimson in receptor neurons for $CO_2$ or for bitter taste (using the indicated drivers and *20xUAS-CsChrimson-mVenus in attP18*) avoided the illuminated quadrants, whereas the control genotype (empty driver, *pBDPGAL4U in attP2/20xUAS-CsChrimson-mVenus in attP18*) showed a very slight preference for illuminated quadrants. Lighter colored areas around lines indicate the standard error of the mean. (**C**) Screening of MBON drivers for behavioral valence. The red-light preference index (PI) was defined as: [(number of flies in illuminated quadrants) - (number of flies in non-illuminated quadrants)]/(total number of flies). For each 2-min experiment, the overall PI was calculated by averaging the PIs from the final 5 s of each of the two red-lights-on conditions (namely, 55–60 s and 115–120 s). MBON cell types expressed in each driver are shown in the matrix below the graph; collectively all the drivers covered 20 of the 22 MBON types. MBON-γ3 and MBON-γ3β'1 are listed together, because these MBON cell types are always labeled together in our split-GAL4 lines. Driver lines shown in bold were selected for more detailed experiments. MBONs have been grouped by cell body cluster, neurotransmitter and color-coded as indicated (see *Table 1*). The bottom and top of each box represents the first and third quartile, and the horizontal line dividing the box is the median. The whiskers represent the 10[th] and 90[th] percentiles. The gray rectangle spanning the horizontal axis indicates the interquartile range of the control. Relative expression levels in individual cell types were estimated by confocal microscopy of brains stained for CsChrimson-mVenus and are shown in gray scale (See 'Materials and methods'; *Figure 3* and *Figure 3—figure supplements 1,2* show confocal images). Statistical tests are described in methods: *, p < 0.05; **, p < 0.01; ***, p < 0.001; n = 11–29 for GAL4/CsChrimson; n = 67 for control.

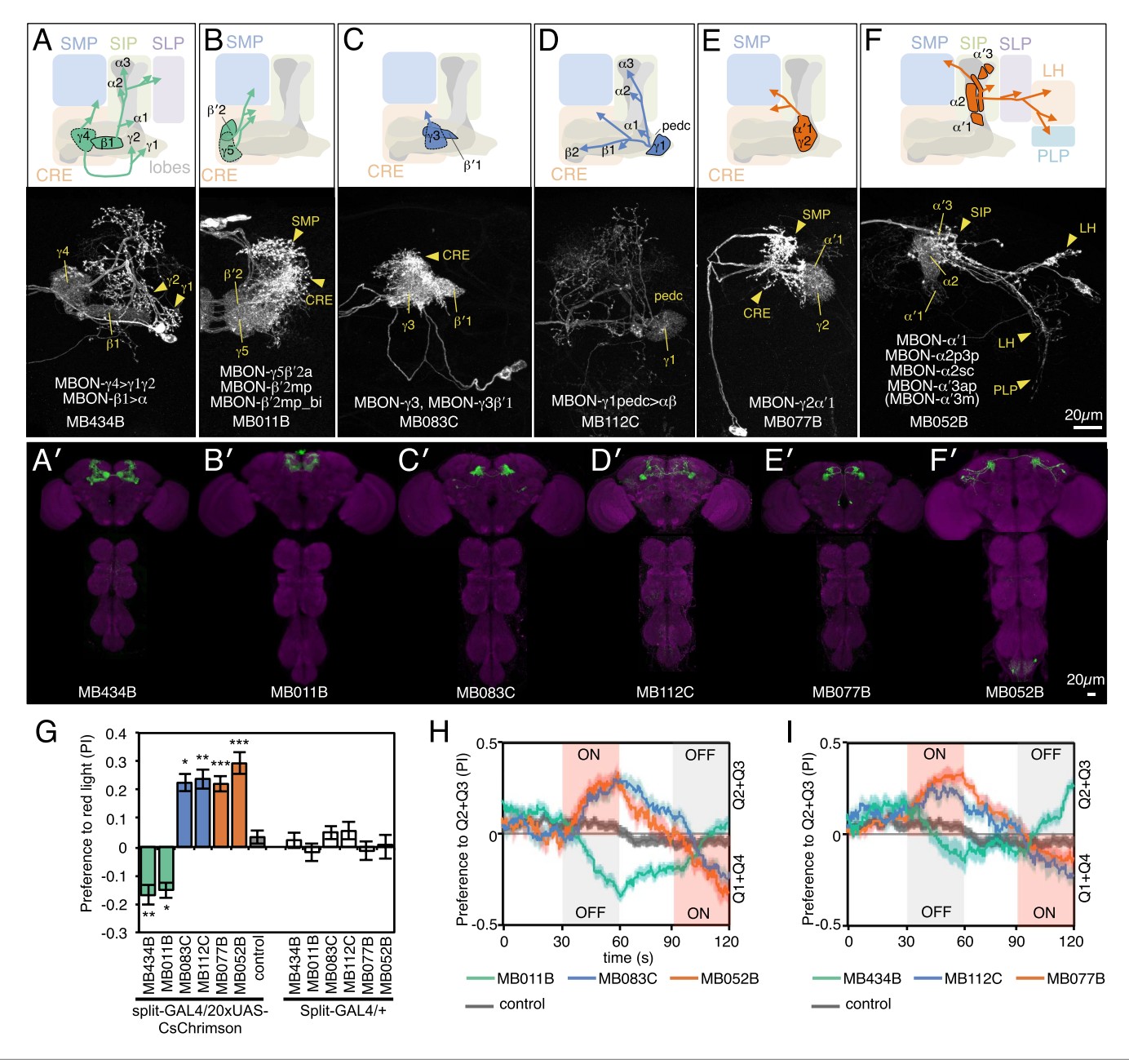

**Figure 3**. MBONs for attraction and repulsion. (**A–F**) Anatomy of MBONs in six selected drivers. Top: Diagrams of the cell types that show expression with each driver, color-coded by neurotransmitter. The name of the split-GAL4 driver line and the cell type(s) in which it expresses are given at the bottom of each panel. The positions of their dendrites in the MB lobes are indicated; the arrows show the projections of their axons. Bottom: Confocal images of a single brain hemisphere for each line; MB compartments occupied by the dendrites of the MBONs in each line are labeled and the arrowheads indicate the sites of their synaptic terminals. See *Figure 1* legend for abbreviations. Full confocal stacks of these images are available at www.janelia.org/split-gal4. (**A′–F′**) Expression patterns of the same drivers shown in a complete brain and ventral nerve cord: green, expression of CsChrimson-mVenus; magenta, neuropil reference stain (nc82 antibody). Frontal views of maximum intensity projections are shown. See the accompanying manuscript (*Aso et al., 2014*) for a detailed description of the morphology of each cell type. (**G**) The preference index (PI) for each experimental group (split-GAL4/CsChrimson) was compared with split-GAL4/+ and *pBDPGAL4U*/CsChrimson controls. Bars and error bars indicate mean and standard error of the mean respectively. Asterisk indicates significance: *, $p < 0.05$; **, $p < 0.01$; ***, $p < 0.001$. (**H–I**) Time course of the preference index. See legend to *Figure 2* for more explanation. Data for six drivers and control are displayed in two panels for clarity, color-coded as indicated.

*Figure 3. Continued on next page*

*Figure 3. Continued*

The following figure supplements are available for figure 3:

**Figure supplement 1**. Expression patterns of split-GAL4s with UAS-CsChrimson.

**Figure supplement 2**. Expression patterns of split-GAL4s with UAS-CsChrimson.

showed significant preference to the red light in the absence of the CsChrimson effector (*Figure 3G*), confirming that phototaxis was limited at the wavelength and intensity of light used. These results demonstrate that activation of MBONs is itself sufficient to elicit either approach or avoidance, depending on the cell type.

Although activation of individual cell types (see *Figure 3D,E*) can result in robust phenotypes, some of the strongest effects were observed with drivers that express in combinations of MBONs. For example, MB011B and MB052B, which drive expression in groups of three or five cell types that use the same transmitter (the glutamatergic M4/M6 and cholinergic V2 clusters, respectively; *Table 1*), caused strong responses, while activation with drivers for smaller subsets or individual cell types within these groups had much reduced effects (*Figure 2C*). Although activation of the single cell types MBON-γ1pedc>α/β or MBON-γ4>γ1γ2 had a significant effect, these neurons send axonal projections to other compartments in the MB lobes, giving them the potential to directly influence the activity of additional MBON cell types. Given that multiple MBONs can independently contribute to behavioral valence as measured by attraction vs repulsion, we sought to determine how conflicting or consonant information from multiple MBONs is integrated to bias behavior. We combined two split-GAL4 drivers for cell types with either similar or opposite effects in the same fly (*Figure 4*, *Figure 4—figure supplement 1*; 'Materials and methods'). When combining drivers eliciting similar responses, flies generally showed a stronger response than to either driver alone. Conversely, co-activation of MBON cell types with opposing effects resulted in intermediate responses. Together, these data are consistent with a simple combinatorial model of valence integration.

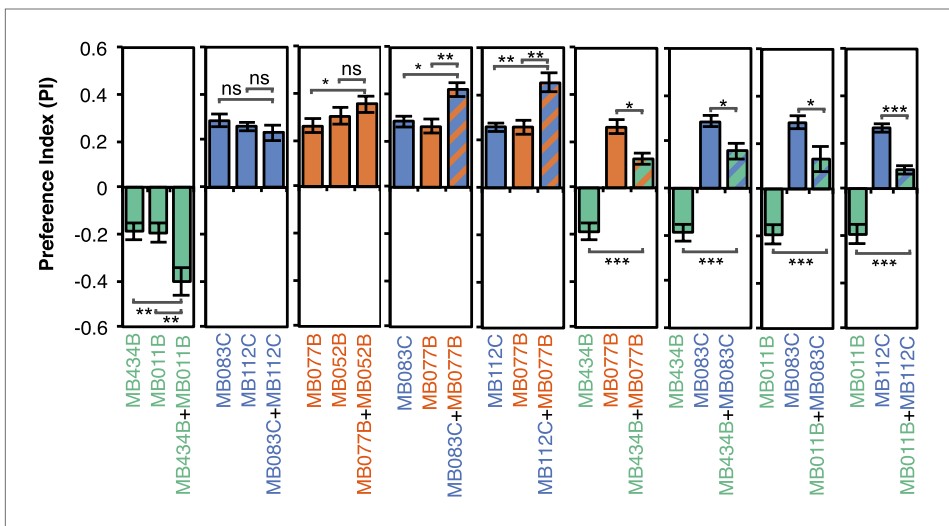

**Figure 4**. Additive effects of MBONs for attraction and repulsion. PIs of individual drivers and combinations of drivers are shown when tested by the protocol shown in *Figure 2A*. The expression patterns of the combination drivers are shown in *Figure 4—figure supplement 1*. Bar graphs are color coded by the transmitter of the MBONs. To facilitate comparisons, data for some genotypes are shown in more than one panel. Asterisk indicates significance: *, $p < 0.05$; **, $p < 0.01$; ***, $p < 0.001$.

The following figure supplement is available for figure 4:

**Figure supplement 1**. Expression patterns of split-GAL4 combinations.

How do MBONs bias behavior to cause approach or avoidance? A conditioned response to a learned odor is unlikely to be achieved by eliciting a predetermined motor pattern. To carry out appropriate changes in speed and direction, a fly needs to evaluate both the valence of the learned odor and its own trajectory relative to the location of the odor source. To assess which aspects of locomotion are modified by MBON activation to generate a bias between approach and avoidance, we analyzed the behavior of flies in a 10 mm-wide graded lighting choice zone centered on the border between dark and illuminated quadrants. We tracked the trajectories of individual flies (*Figure 5A–C*) and calculated the fraction of flies crossing the choice zone from the 'light-off' to the 'light-on' area (and vice versa) as well as the fraction of flies changing direction in the choice zone thus returning to the area they came from.

Most control flies entering the choice zone from either the light-off or light-on side continued moving forward, crossing into the other side (*Figure 5D,E*; *Video 1*). Flies expressing CsChrimson in either GABAergic (MB112C or MB083) or cholinergic (MB077B or MB052B) MBONs behaved similarly to control flies when entering the choice zone from the light-off area. When they entered from the light-on side, these flies showed a slight tendency to turn around in the choice zone (*Figure 5D*) consistent with their preference for illuminated areas (*Figure 2C*). Flies expressing CsChrimson in a combination of GABAergic and cholinergic MBONs (MB112C plus MB077B), which displayed the strongest preference for lighted areas (*Figure 4*), also showed the highest rates of exiting to the light-on side when entering the choice zone from the light-off area (*Figure 5D*). On the other hand, flies expressing CsChrimson in glutamatergic MBONs (MB434B, MB011B or a combination of them) frequently turned around in the choice zone when entering from the light-off side while crossing into the light-off area when entering the choice zone from the illuminated side (*Figure 5D,E*; *Video 2* and *Video 3*), behaviors that are consistent with these flies' avoidance of illuminated areas (*Figure 2C*).

We next calculated the fraction of flies exiting the choice zone into the light-on side irrespective of its entry direction and found this 'choice probability' to be highly correlated with preference index (*Figure 5F*). In contrast, we found little correlation between preference index and either mean walking speed or mean angular velocity, an indicator of turning probability, of flies in illuminated quadrants (*Figure 5G,H*). Flies are able to adjust these parameters to execute avoidance behaviors in other contexts; we found that activation of some chemosensory neurons and projection neuron combinations that repelled flies significantly altered both walking and angular speed (*Figure 5—figure supplement 1*; *Video 4* and *Video 5*). Our results indicate that MBON activity biases the direction that flies turn in the choice zone, thereby biasing the direction in which they exit that zone. We did not observe a stereotyped turning behavior in the choice zones; more specifically, the time between entering the choice zone and making a turn as well as the precise direction of the turn varied (*Figure 5E*; *Videos 2 and 3*). Moreover, flies displayed apparently normal behavior within the uniform illumination of lighted quadrants (*Video 2 and 3*), showing no apparent increase in their speed and turning rates. These observations support the view that MBONs represent valence, abstract information that serves to bias—rather than direct—specific motor patterns.

## Requirement of MBONs in two-hour odor-shock and odor-sugar associative memory

We asked which MBONs were required for aversive and appetitive odor memory, using a well-established discriminative olfactory learning paradigm (*Figure 6A*) (*Tempel et al., 1983*; *Tully and Quinn, 1985*; *Schwaerzel et al., 2003*; *Gerber et al., 2004*; *Davis, 2005*). In this paradigm, flies are exposed to an odor together with an unconditioned stimulus (US) of either an electric shock punishment or a sugar reward, and then to another odor without the US. The 'trained' flies are tested at a later time to determine if they exhibit a differential response to the two odors, which is taken as indication of memory formation.

In the first set of experiments we assayed memory 2 hr after training, using a set of 23 split-GAL4 lines to transiently block neuronal activity in different subsets of MBON cell types. Two memory processes that are thought to rely on different molecular and circuit mechanisms, anesthesia resistant memory and anesthesia sensitive memory, contribute to memory at this retention time (*Dudai et al., 1976*; *Folkers et al., 1993*; *Isabel et al., 2004*; *Krashes and Waddell, 2008*; *Aso et al., 2010*; *Pitman et al., 2011*; *Knapek et al., 2011*). We blocked neuronal function throughout the training, retention and test periods. Thus we expect to detect impairments in any phase of memory processing, including formation, consolidation and retrieval.

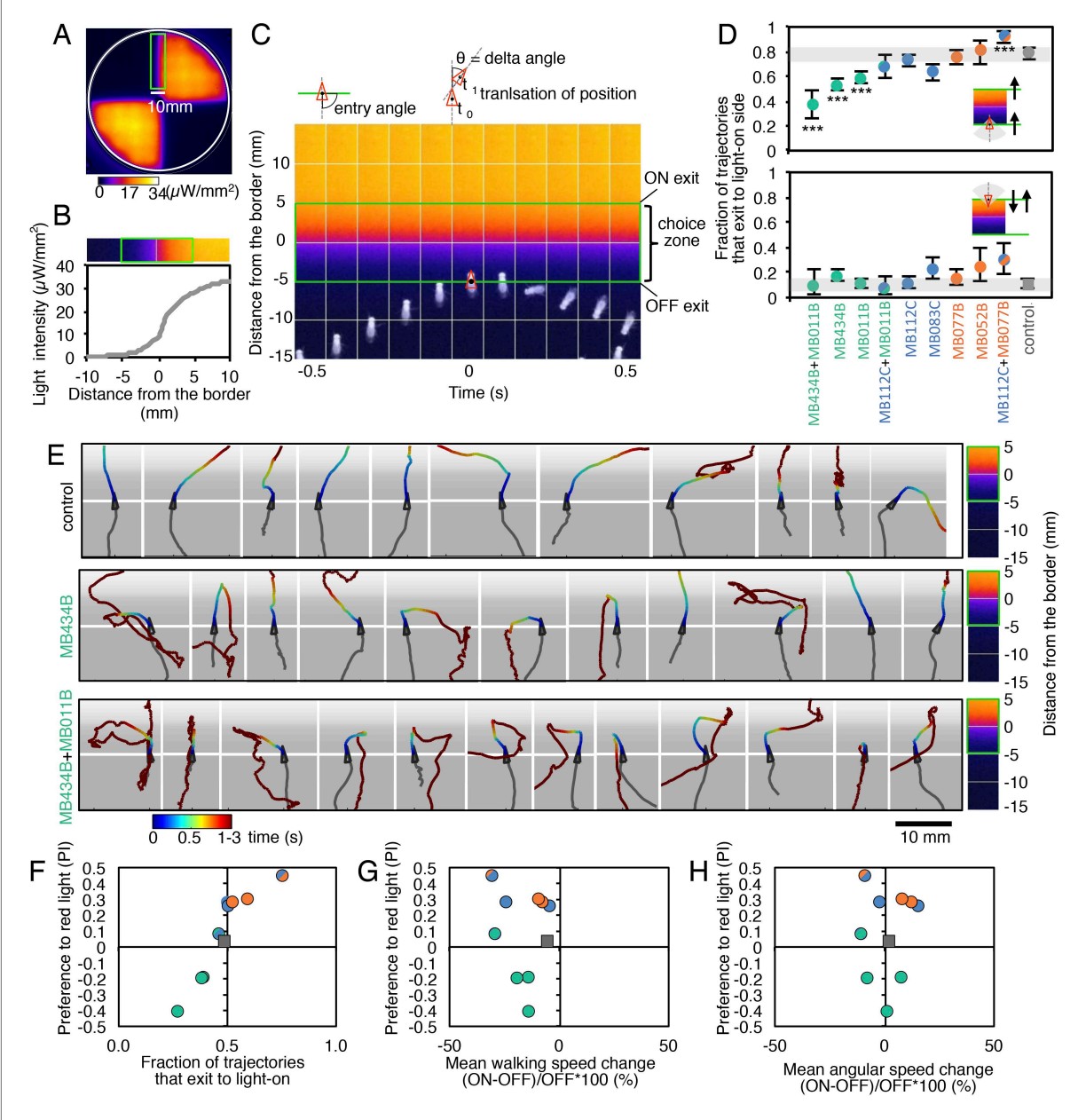

**Figure 5**. MBONs bias choice at the border. (**A**) Pseudo-colored light intensity in the behavioral arena. The choice zone was defined as ± 5 mm from the light ON/OFF border (green box). (**B**) Gradient of light intensity across the light ON/OFF border axis shown. (**C**) An example of MB434B + MB011B/UAS-CsChrimson fly that entered the choice zone from the light-off side at time = 0 and then stopped and quickly turned around to exit the choice zone, going back into the light-off side. Eleven images taken at 0.1 s intervals have been superimposed. Entry angle to the choice zone was defined as diagramed (top left). Speed and angular speed was calculated based on the difference in the position of a fly in successive frames of 30 frames per second video recordings. (**D**) The fraction of trajectories entering the choice zone from the dark side and then exiting to the illuminated side is plotted for the indicated drivers in combination with CsChrimson (top). The control genotype was the empty driver, *pBDPGAL4U,* in *attP2* in combination with *20xUAS-CsChrimson-mVenus in attP18*. Only flies that entered the choice zone at an entry angle of between 45 and 135° (facing to the light ON/OFF border) and had moved more than 5 mm in the 1 s prior to entering the choice zone were analyzed. The error bars show the 95% confidence interval. Between 79 and 410 trajectories were analyzed per genotype. Compared to the control, MB434B + MB011B, MB434B and MB011B showed a significantly lower fraction of trajectories that exit to the light-on side (more avoidance of light), whereas significantly higher fraction of trajectories of MB112C + MB077B flies exit to the light-on side (more attraction to the light); multiple comparisons with the Dunn-Sidak correction: ***, p < 0.001. Similarly, the fraction of trajectories entering the choice zone from the illuminated side and then changing direction so as to also exit to the illuminated side is plotted for the indicated drivers in combination with CsChrimson (bottom). Between 43 and 280 trajectories were analyzed per genotype. (**E**) Representative trajectories

*Figure 5. Continued on next page*

**Figure 5. Continued**

are shown for the indicated genotypes. The trajectories are color-coded to indicate the position of the fly in the trajectory as a function of time after entering the choice zone. The triangle shows the position of the fly at time = 0, when flies entered into the choice zone (indicated by the white line at −5 mm from the light ON/OFF border). The gray scale background in the panels and pseudo-color scale on the right indicate the intensity of CsChrimson activating light. (**F**) Preference index to the CsChrimson-activating light (**Figure 4**) was plotted against the fraction of trajectories that exit from the choice zone to the illuminated side irrespective of side of entry: [(number of trajectories enter to the choice zone from dark side and then exit to illuminated side) + (number of trajectories enter to the choice zone from illuminated side and then exit to illuminated side)] divided by total number of trajectories that entered into the choice zone. They were highly correlated (Spearman's rank-order correlation: Pearson r = 0.91; R square = 0.83; p < 0.001). Genotypes are the same as in panel D and are shown with same color code. (**G**) Preference index to the CsChrimson light (**Figure 4**) was plotted against the mean walking speed change in the illuminated quadrants compared to dark quadrants (see **Figure 5—figure supplement 1**). There was no significant correlation (Spearman's rank-order correlation: Pearson r = 0.13; R square = 0.01; p = 0.72). Genotypes are the same as in panel D and are shown with same color code. (**H**) Preference index to the CsChrimson light (**Figure 4**) was plotted against mean angular speed in the illuminated quadrants compared to dark quadrants (see **Figure 5—figure supplement 1**). There was no significant correlation (Spearman's rank-order correlation: Pearson r = 0.12; R square = 0.001; p = 0.75). Genotypes are the same as in panel D and are shown with same color code.

The following figure supplement is available for figure 5:

**Figure supplement 1**. MBON activation has only small effects on walking and angular speeds.

We first screened the lines using a strong Shibire[ts1] effector (*pJFRC100-20XUAS-TTS-Shibire-ts1-p10 in VK00005*). Because some lines had phenotypes at the permissive temperature, presumably due to the effector's high level of expression, we retested the nine lines that showed a reduction in memory performance with a weaker Shibire[ts1] effector (*UAS-Shi x1*; see **Figure 6** legend and 'Materials and methods'). With these parameters, only one line, MB112C, also showed significant memory impairment (**Figure 6B**). The experimental MB112C flies showed significantly lower memory performance than genetic control groups at the restrictive temperature (**Figure 6C**), but not at the permissive temperature (**Figure 6D**). These flies displayed normal shock and odor avoidance at the restrictive temperature (**Figure 6E,F**), indicating that the observed memory impairment was not due to a defect in sensory or motor pathways. MB112C drives expression in MBON-γ1pedc>α/β (**Figure 6—figure supplement 1A**). We confirmed the requirement for this cell type using a second driver, MB085C, not included in the original 23 lines screened (**Figure 6C**; **Figure 6—figure supplement 1A**).

One of the PPL1 cluster dopaminergic neurons, PPL1-γ1pedc (also known as MB-MP1), innervates the same MB compartments as MBON-γ1pedc>α/β. Blocking PPL1-γ1pedc activity, using the MB438B split-GAL4 driver, also impaired aversive memory (**Figure 6C–F**). Conversely, activation of PPL1-γ1pedc with the temperature gated cation channel dTrpA1 (**Hamada et al., 2008**) substituted for electric shock as the unconditioned stimulus (US), inducing robust aversive odor memory (**Figure 6G**). This confirms a conclusion reached using less specific enhancer trap GAL4 drivers (**Aso et al., 2010, 2012**). Consistent with these observations, restoring expression of the D1-like dopamine receptor specifically in the γ Kenyon cells has been shown to rescue the aversive odor memory defect of a receptor mutant (**Qin et al., 2012**). MBON-γ1pedc>α/β is immunoreactive to GABA (**Figure 6—figure supplement 1C**) and is one of only three MBON cell types with axon terminals within the MB lobes (**Figure 6H**; **Figure 6—figure supplement 1D,E**) (**Aso et al., 2014**). We visualized the single-cell morphologies of MBON-γ1pedc>α/β (**Figure 6I**) and PPL1-γ1pedc (**Figure 6J**) and using two-color labeling confirmed that the axon terminals of the DAN (one or two cells per brain hemisphere) precisely overlap with the dendrites of the MBON (a single cell per brain hemisphere) in the MB pedunculus and γ1 compartment (**Figure 6K**). This establishes two essential components of a circuit for 2-hr aversive odor memory.

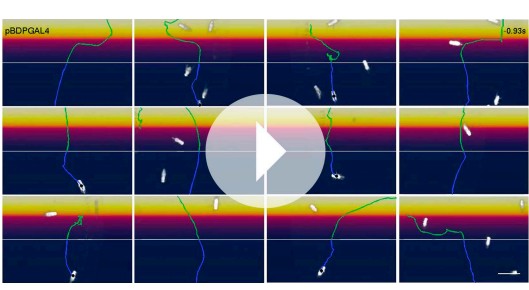

**Video 1**. Choice behaviors of control flies.
A representative 12 trajectories for the control genotype (pBDPGAL4U/CsChrimson) showing flies that entered the choice zone from the light-off side at the 1-s time point and their subsequent behavior for 3 s. See legend of **Figure 5E** for more explanation. Time is shown in the upper right corner.

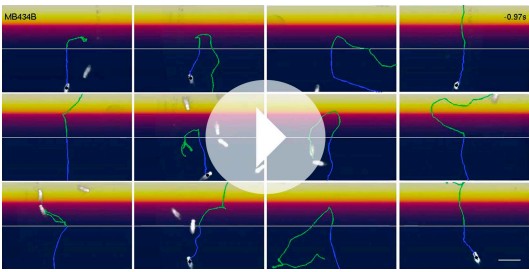

**Video 2**. Choice behaviors of MB434B/CsChrimson flies. A representative 12 trajectories for MB434B/CsChrimson flies showing flies that entered the choice zone from the light-off side at the 1-s time point and their subsequent behavior for 3 s. When two glutamatergic MBONs (MBON-γ4>γ1γ2 and MBON-β1>α) were activated using MB434B/CsChrimson, flies tended to avoid entering light-on quadrants. The motor patterns they used for avoiding the light-on quadrants were not stereotyped, as illustrated by the randomly selected examples shown in this video.

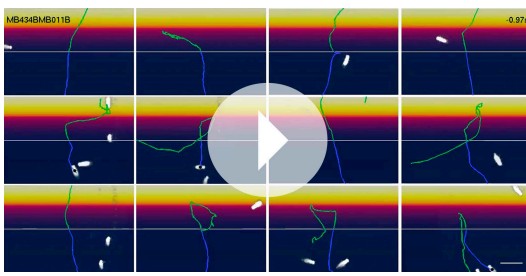

**Video 3**. Choice behaviors of MB434B + MB011B/CsChrimson flies. A representative 12 trajectories for MB434B + MB011B/CsChrimson flies showing flies that entered the choice zone from the light-off side at the 1-s time point and their subsequent behavior for 3 s. When five glutamatergic MBONs (MBON-γ4>γ1γ2, MBON-β1>α, MBON-γ5β'2a, MBON-β'2mp and MBON-β'2mp_bilateral) were activated using MB434B + MB011B/CsChrimson, the avoidance response is observed in a slightly higher fraction of flies than in *Video 2*.

A larger set of MBONs was involved in 2-hr appetitive olfactory memory (*Figure 7*). Inactivation of the neurons represented in 12 out of the 23 split-GAL4 lines tested showed an effect in the initial screening with the strong Shibire[ts1] effector; when retested with the weaker effector, eight of these lines produced a significant impairment (*Figure 7*, *Figure 7—Figure supplement 1*). All eight lines displayed normal memory at the permissive temperature (*Figure 7C*) and normal attraction to sugar at the restrictive temperature (*Figure 7D*). Five of these lines (MB082C, MB093C, MB018B, MB051B and MB077B) express in subsets of the so-called V3/V4 cluster of cholinergic MBONs (MBON-γ2α'1, MBON-α'2 and MBON-α3; *Table 1*), establishing a role for this group of MBONs in 2-hr appetitive memory. MB310C labels the glutamatergic MBON-α1. MB011B labels three types of M4/M6 cluster glutamatergic MBONs (MBON-γ5β'2a, MBON-β'2mp and MBON-β'2mp_bilateral; *Table 1*), suggesting a role for one or more of these cell types. Two lines that express in subsets of MB011B cell types, MB210B and MB002B, showed a memory defect in the primary screening but failed to reach statistical significance when retested with the weaker effector. Similarly, we observed that activating CsChrimson with MB011B, but not with MB210B or MB002B, produced a significant aversive effect (see *Figure 2C*). These data are consistent with an additive role of these MBONs on behavior, which is further supported by the anatomical observation that the axon terminals of some of these MBONs converge to the same areas outside the MB (*Figure 7E,F*) (*Aso et al., 2014*).

## Requirement of the MB output neurons in visual memory

The MB has been proposed to play diverse roles in visual behaviors including context generalization and sensory preconditioning between olfactory and visual cues (*Liu et al., 1999*; *Zhang et al., 2013, 2007*; *Brembs, 2009*; *van Swinderen et al., 2009*). Using visual learning assays in which flies are trained to associate a color (blue or green) with either an electric shock punishment or a sugar reward (*Figure 8A* and *Figure 9A*), *Vogt et al., 2014* demonstrated that γ-lobe KCs are required for immediate visual associative memory and that activation of specific subsets of PPL1 and PAM cluster DANs can substitute as the US for electric shock or sugar, respectively. Here we use the same assays to ask which MBONs are required for visual memory. We used the strong Shi[ts] effector (*pJFRC100-20XUAS-TTS-Shibire-ts1-p10 in VK00005*) to silence MBONs (Materials and methods).

For aversive visual memory (*Figure 8B*), we identified one driver line, MB112C, that labels MBON-γ1pedc>α/β. We confirmed the requirement for this cell type with two additional split-GAL4 lines, MB262B (*Figure 8C*; *Figure 8—figure supplement 1*) and MB085C (data not shown). This is the same MBON we found to be important for olfactory aversive memory (*Figure 6*) and so we asked if the same DAN as in olfactory memory was likewise required. Indeed, we found that MB438B, a driver line for PPL1-γ1pedc, showed a significant impairment (*Figure 8C*; *Figure 8—figure supplement 1*).

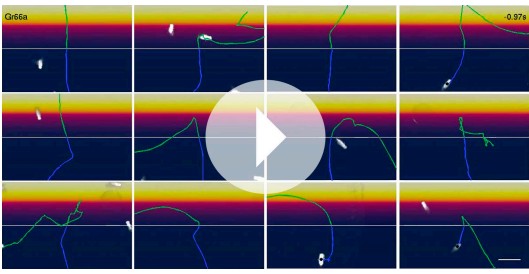

**Video 4**. Choice behaviors of Gr66a-GAL4/CsChrimson flies. A representative 12 trajectories for Gr66a-GAL4/CsChrimson flies showing flies that entered the choice zone from the light-off side at the 1-s time point and their subsequent behavior for 3 s. When the Gr66a sensory neurons for bitter taste were activated using Gr66a-GAL4/CsChrimson, flies increased their velocity when they entered the illuminated area, which facilitated their escape from light-on quadrants (see **Figure 2B**).

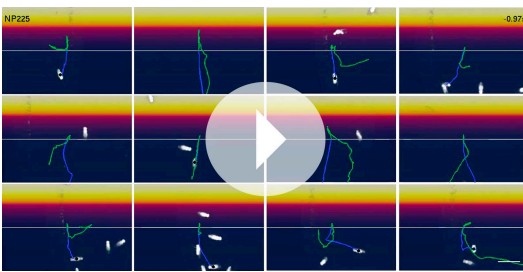

**Video 5**. Choice behaviors of NP225-GAL4/CsChrimson flies. A representative 12 trajectories for NP225-GAL4/CsChrimson flies showing flies that entered the choice zone from the light-off side at the 1-s time point and their subsequent behavior for 3 s. When this broad set of antennal lobe projection neurons was activated using NP225-GAL4/CsChrimson, flies showed a stereotyped backward walking behavior when they approached the illuminated area; a response was observed at lower light intensities than with the other driver lines. Flies that were already in the illuminated quadrants when the light turned on showed continuous rotation that typically lasted for entire light-on period (30 s) and often extended a few seconds after the red light was turned off (not shown).

The experimental MB112C, MB262B and MB438B lines showed normal memory at the permissive temperature (**Figure 8C**) and normal shock avoidance at the restrictive temperature (**Figure 8D**). Our finding that the same MBON and its modulatory DAN were required for both visual and olfactory aversive memory suggests that this local MB circuit is important for aversive memory in general, rather than specifically for a particular sensory modality.

For appetitive visual memory, we identified five drivers with significant impairments: MB434B, MB011B, MB210B, MB542B and MB052B (**Figure 9B,C**; **Figure 9—figure supplement 1**). These lines showed normal memory at the restrictive temperature in the absence of the effector (**Figure 9C**) and at the permissive temperature with the effector (**Figure 9D**), as well as normal sugar attraction at restrictive temperature (**Figure 9E**). The anatomy of the cell types in these driver lines is illustrated in **Figure 9F,G**. Unlike in the case of aversive memory, the cell types required for appetitive memory differed somewhat between visual and olfactory modalities. Nevertheless, these modalities both employed the M4/M6 cluster glutamatergic MBONs labeled in MB011B and MB210B (MBON-γ5β′2a, MBON-β′2mp and MBON-β′2mp_bilateral; **Table 1**). Moreover, the observation that multiple cholinergic and glutamatergic MBONs play a role in appetitive memory, but not aversive memory, was shared across modalities.

## Requirement of MBONs in retrieval of long-term odor-shock memory

Repeated pairing of an odor and an electric shock, with inter-trial rest intervals, results in protein synthesis dependent aversive long-term memory (LTM) (**Tully et al., 1994**). The molecular and cellular mechanisms underlying aversive LTM are known to differ from those responsible for memories with shorter retention times (**Yin et al., 1994**; **Pascual and Preat, 2001**; **Dubnau et al., 2003**; **Comas et al., 2004**; **Isabel et al., 2004**; **Yu et al., 2006**; **Blum et al., 2009**; **Akalal et al., 2010, 2011**; **Trannoy et al., 2011**; **Huang et al., 2012**; **Placais et al., 2012**).

We assessed the requirement of MBONs in the retrieval phase of aversive LTM by training flies at permissive temperature and blocking their activity only during the memory test at 24 hr after training (**Figure 10A**). Inactivation of MB052B, which broadly labels cholinergic V2 cluster MBONs (**Figure 3F**; **Table 1**), resulted in nearly complete loss of aversive odor LTM recall (**Figure 10C**). Our results confirm a previously reported requirement for the V2 cluster MBONs in long-term aversive odor memory recall (**Sejourne et al., 2011**). We also assayed five lines that express in subsets of the MBONs found in MB052B. While some of them showed lower memory scores, none showed a statistically-significant memory impairment (**Figure 10B**). Therefore, our results suggest that V2 cluster MBONs (**Figure 10D**) function as a group in the retrieval of aversive odor LTM. Consistent

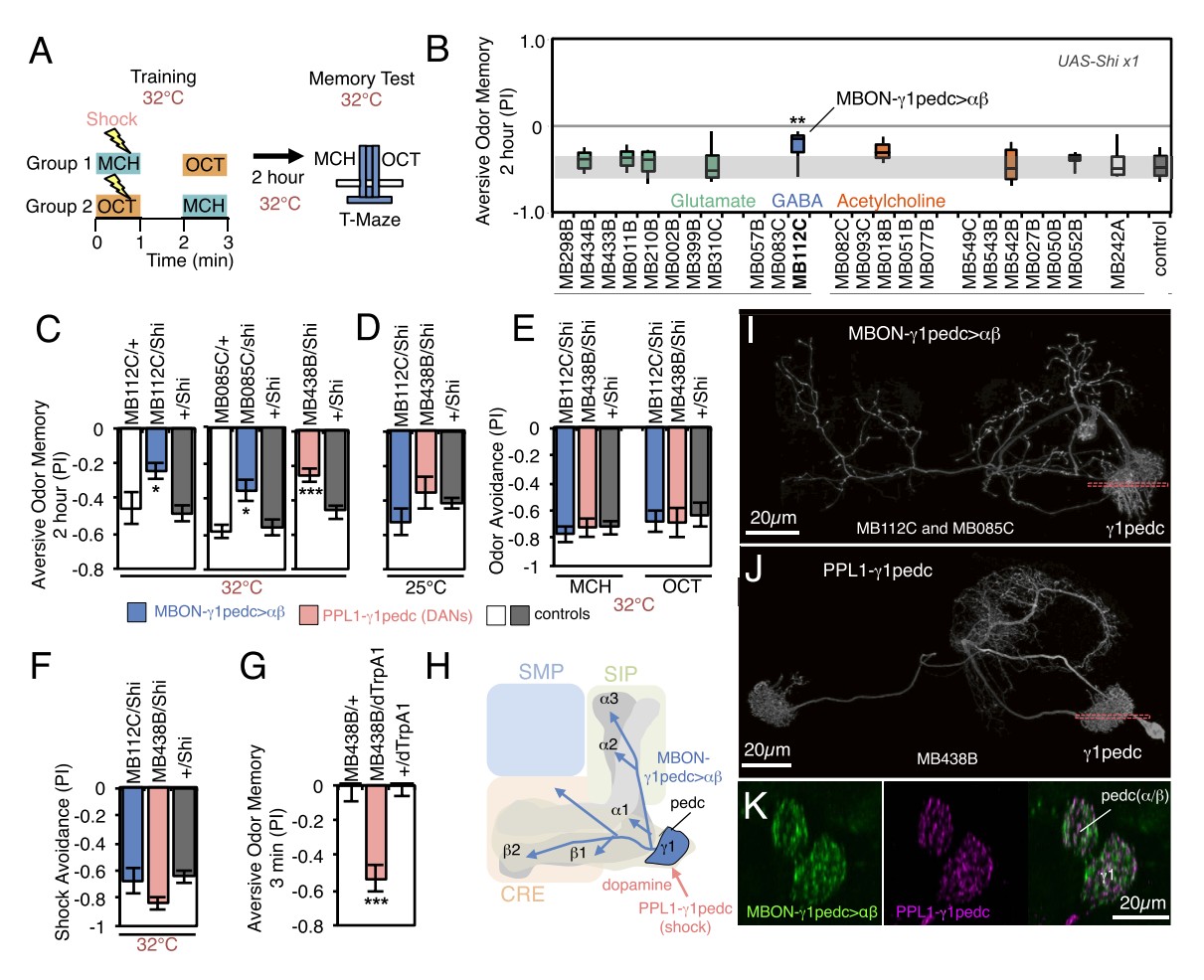

**Figure 6**. Requirement of MBONs for 2-hr aversive odor memory. (**A**) Schematic of the T-maze apparatus. In one group of flies, 4-methylcyclohexanol (MCH) was paired with electric shocks for 60 s. After a 60 s pause, 3-octanol (OCT) was delivered without electric shock. For the reciprocal group, OCT was paired with electric shock, and MCH was delivered without shock. 2 hr after training, flies were given a choice between the two odors in a T-maze and their distribution was used to calculate a performance index (PI). The PI corresponds to the mean of the [(number of flies in the OCT tube minus number of flies in the MCH tube)/total number of flies when OCT was paired with electric shock] and [(number of flies in the MCH tube minus number of flies in the OCT tube)/total number of flies when MCH was paired with electric shock]. The set of 23 driver lines was first screened using *pJFRC100-20XUAS-TTS-Shibire-ts1-p10* (*VK00005*) as the Shi[ts] effector; lines that showed a strongly decreased PI in this initial screen were retested with the *UAS-Shi x1* effector. (**B**) Results of secondary screening of MBONs for 2-hr aversive memory using *UAS-Shi x1*. MBONs have been grouped by neurotransmitter and color-coded. The bottom and top of each box represents the first and third quartile, and the horizontal line dividing the box is the median. The whiskers represent the 10[th] and 90[th] percentiles. The gray rectangle spanning the horizontal axis indicates the interquartile range of the control. Statistical tests are described in methods: **, p < 0.01. (**C**) 2-hr aversive odor memory at the restrictive temperature. In addition to confirming the result obtained with MB112C in the screening assays, a second driver line for MBON-γ1pedc>α/β, (MB085C) was tested and found to have a similar impairment. Blocking dopaminergic input to the MB compartments where the dendrites of MBON-γ1pedc>α/β reside, using MB438B, also impairs 2-hr aversive odor memory. *UAS-Shi x1* was used as the effector. See ***Figure 6—figure supplement 1*** for the expression patterns of the MB112C, MB085C and MB438B drivers. Bars and error bars show mean and standard error of the mean (SEM). Statistical tests are described in methods: *, p < 0.05; ***, p < 0.001. (**D**) 2-hr aversive odor memory in MB112C/Shi and MB438B/Shi flies is not impaired at the permissive temperature. *UAS-Shi x1* was used as the effector. (**E**) Avoidance of MCH and OCT in untrained MB112C/Shi and MB438B/Shi flies is not impaired at the restrictive temperature, indicating that these genotypes can detect and respond to the odors. *UAS-Shi x1* was used as the effector. (**F**) Shock avoidance in untrained MB112C/Shi and MB438B/Shi flies is not impaired at the restrictive temperature. *UAS-Shi x1* was used as the effector. (**G**) Activation of the PPL1-γ1pedc dopaminergic neurons can substitute for electric shock as the unconditioned stimulus (US). Flies trained using thermoactivation of MB438B/dTrpA1 as the US showed robust 3 min aversive memory. (**H**) Diagram of a circuit module for aversive odor memory. MBON-γ1pedc>α/β is shown in blue. PPL1-γ1pedc conveys the US of electric shock (pink arrow). (**I**) Morphology of MBON-γ1pedc>α/β. A frontal view of maximum intensity projection of an image of a single cell generated by stochastic labeling using the multicolor flip-out technique (MCFO; Nern et al., in prep.) is shown. MBON-γ1pedc>α/β has dendrites in γ1 and the core of pedunclulus (pedc), where α/β KCs project. Its axon bilaterally innervates the α/β lobes and the pedc, and neighboring neuropils

*Figure 6. Continued on next page*

*Figure 6. Continued*

(SMP, SIP and CRE); see *Figure 6—figure supplement 1* for more anatomical details. The orange dashed line indicates the plane of the cross section shown in (**K**) of the γ1 and the pedc. (**J**) Morphology of PPL1-γ1pedc. A frontal view of maximum intensity projection of an image of a single cell generated by MCFO is shown. PPL1-γ1pedc extends dendrites unilaterally in the SMP, SIP and CRE, and bilaterally innervates the γ1 and the pedc. The orange dashed line indicates the plane of the cross section of γ1 and the pedc shown in (**K**). (**K**) Double labeling of MBON-γ1pedc>α/β (green; *R12G04-LexA, pJFRC216-13XLexAop2-IVS-myr::smGFP-V5* [*su(Hw)attP8*] and PPL1-γ1pedc (magenta; MB320C, *pJFRC200-10XUAS-IVS-myr::smGFP-HA* (*attP18*)). A coronal section is shown. The dendrites of MBON-γ1pedc>α/β and the terminals of PPL1-γ1pedc precisely overlap in γ1 and the pedc.

The following figure supplement is available for figure 6:

**Figure supplement 1**. Anatomy of MBON-γ1pedc>α/β and PPL1-γ1pedc, an MBON-DAN pair essential for aversive memory.

with this implied combinatorial action of V2 cluster MBONs, activating subsets of these MBONs with CsChrimson resulted in weak attraction that only reached statistical significance when MB052B was used as the driver (*Figure 2C*). In calcium imaging experiments, V2 cluster MBONs were reported to reduce their response to an odor that had been learned to be aversive (*Sejourne et al., 2011*); this sign of plasticity is consistent with our observation that activation of these neurons elicited attraction.

## Requirement of MBONs in appetitive odor-ethanol intoxication memory

Flies are able to associate an odor with the intoxicating properties of ethanol (*Figure 11A*). Briefly, flies are exposed to two consecutive odors; the second of which is paired with a mildly intoxicating concentration of ethanol vapor. Flies are later tested for their odor preference for the paired vs unpaired odor (*Figure 11A*) (*Kaun et al., 2011*). Flies avoid the odor they experienced at the time of intoxication when tested 30 min after training, but show a long-lasting preference for that odor when tested 24 or more hours later (control data in *Figure 11B,C*) (*Kaun et al., 2011*). Blocking KC synaptic output has been shown to interfere with this memory (*Kaun et al., 2011*), indicating a role for the MB.

To test the role of MBONs in appetitive ethanol memory, we blocked MBON function during training and memory retrieval and assayed for changes in performance. When tested at 24 hr, eight driver lines showed significant memory impairment compared to genetic controls (*Figure 11B,D*; *Figure 11—figure supplement 1*). These eight lines were not significantly different from controls in 30-min aversive memory, although two exhibited a trend towards decreased 30-min memory (*Figure 11C*); the ability to form memories at 30 min establishes that flies of these genotypes can sense the odors and learn to associate them with ethanol.

Our results indicate that MBON-γ4>γ1γ2 (MB434B and MB298B) and MBON-α′2 (MB018B), whose involvement we confirmed with a second line not part of the original screening set (MB091C; *Figure 11E*), are preferentially required for 24 hr memory.

Our data also suggest the involvement of MBON-γ2α′1, where one driver line (MB077B) had a strong effect, while the second driver (MB051B) had a weaker effect that did not reach statistical significance (*Figure 11D*). Our results raise the possibility of the involvement of MBON-α3. However, the two lines we have for this cell type (MB082C and MB093C) gave discordant results and MB082C shows significant expression in MBON-α′2, a cell type that has a large effect on appetitive ethanol memory. Finally, blocking M4/M6 cluster MBONs from γ5 and β′2 (MB011B and MB210B) significantly affected 24 hr memory, but also appeared to decrease 30 min memory. The MBONs required for 24 hr appetitive ethanol memory (*Figure 11F,G*) are partially overlapping with those required for other forms of appetitive memory, again with involvement of multiple glutamatergic and cholinergic MBONs.

Silencing MBON-α′2, a cell type composed of just one cell in each hemisphere (*Figure 11—figure supplement 1F,G*) (*Aso et al., 2014*), resulted in persistence of the aversive memory after 24 hr, when control flies show appetitive memory (see MB018B in *Figure 11D* and MB091C in *Figure 12E*). These results suggest that, while memories for the aversive and rewarding effects of ethanol intoxication are formed simultaneously, they are expressed at different times through independent MB circuits. Moreover, maintenance of the aversive memory upon inactivation of MBON-α′2 argues against a passive dissipation of the aversive memory with time, and implies an active process in the conversion of aversive to appetitive memory.

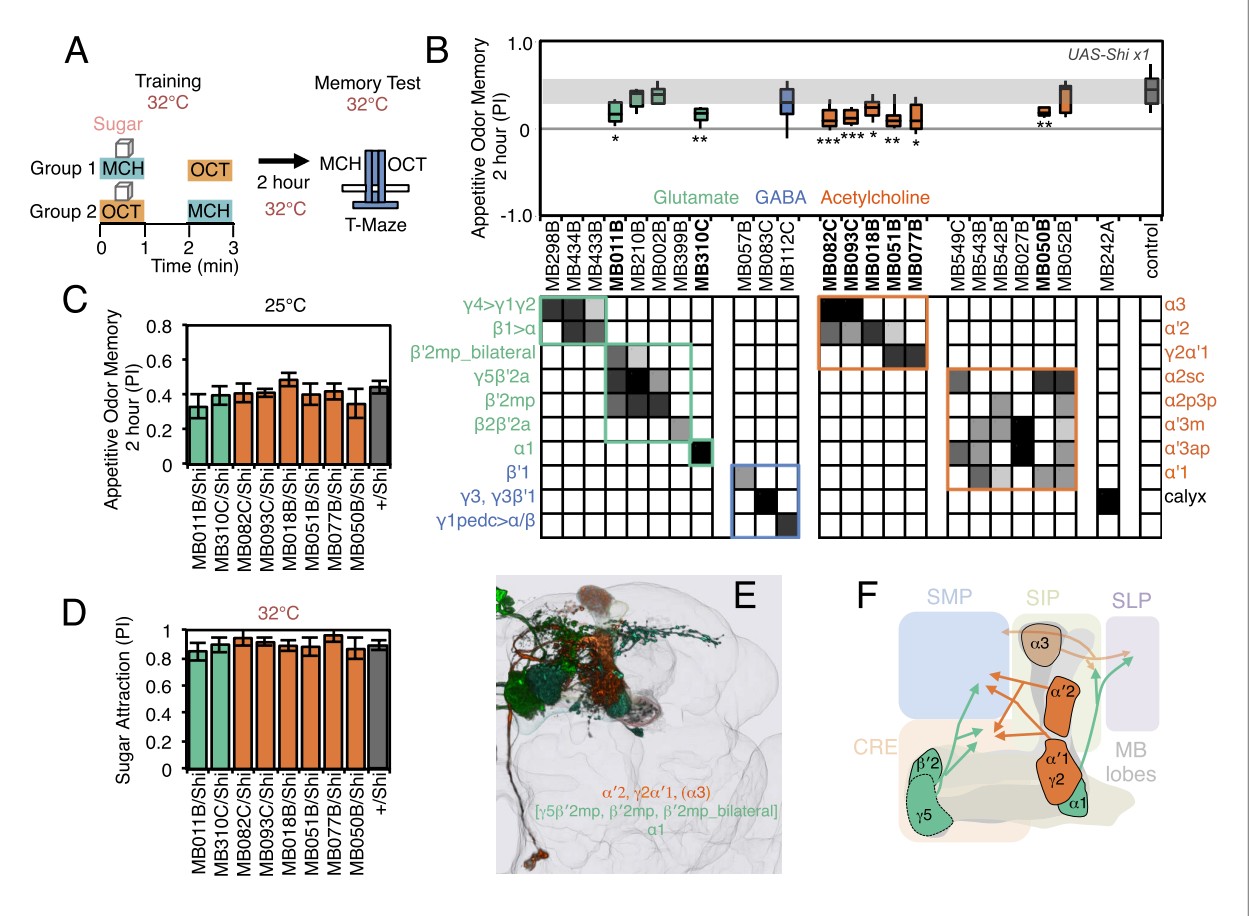

**Figure 7**. Requirement of MBONs for 2-hr appetitive odor memory. (**A**) Flies were trained and tested in a similar way as in **Figure 6A**, except that flies were starved for 28–40 hr prior to experiments and trained with a reward consisting of a tube covered with sugar absorbed filter paper. (**B**) Results of secondary screening of MBONs for 2-hr appetitive memory using *UAS-Shi x1*. MBONs have been grouped by neurotransmitter and color-coded. The bottom and top of each box represents the first and third quartile, and the horizontal line dividing the box is the median. The whiskers represent the 10th and 90th percentiles. The gray rectangle spanning the horizontal axis indicates the interquartile range of the control. Statistical tests are described in methods: *, $p < 0.05$; **, $p < 0.01$; ***, $p < 0.001$. The relative expression levels produced by the split-GAL4 driver lines in each cell type (indicated by the gray scale) are based on imaging with *pJFRC2-10XUAS-IVS-mCD8::GFP (VK00005)*. Images of the expression patterns of selected drivers are shown in *Figure 7—figure supplement 1*. MB082C and MB093C label the cholinergic MBON-α3; however, these lines also show weak labeling in MBON-α'2. As blocking MBON-α'2 alone also gave a significant phenotype (MB018B/Shi), we cannot rule out the formal possibility that the phenotype observed with MB082C and MB093C results from blocking MBON-α'2. MB050B labels MBON-α2 and MBON-α'1; however, because MB052B, which labels the same cell types in addition to others, did not give a phenotype, the requirement for these cell types is not resolved. (**C**) 2 hr appetitive odor memory at permissive temperature. (**D**) Sugar attraction in untrained flies at restrictive temperature. (**E**) Rendering of MBONs with outline of MB lobes and brain. MBONs grouped by square brackets represent cases where the available set of driver lines do not allow assigning an effect to a single cell type, but only to that set of MBONs. MBONs in parentheses represent cases where the data implicating them are only suggestive. (**F**) Diagram of MBONs for appetitive odor memory. MBONs are shown in lighter colors when data implicating them are only suggestive.

The following figure supplement is available for figure 7:

**Figure supplement 1**. Expression patterns of split-GAL4s that caused appetitive odor memory phenotypes.

## MB output neurons bi-directionally regulate sleep

The involvement of the MB in regulating sleep was first established by demonstrating that blocking synaptic output from KCs can either increase or decrease sleep, depending on the GAL4 driver used (*Joiner et al., 2006*; *Pitman et al., 2006*). Sleep in *Drosophila* is under circadian and homeostatic regulation and defined as sustained periods of inactivity (5 min or more) that coincide with increased sensory thresholds, altered brain activity and stereotyped body posture (*Hendricks et al., 2000*; *Shaw et al., 2000*; *Hendricks and Sehgal, 2004*; *Ganguly-Fitzgerald et al., 2006*; *Parisky et al., 2008*;

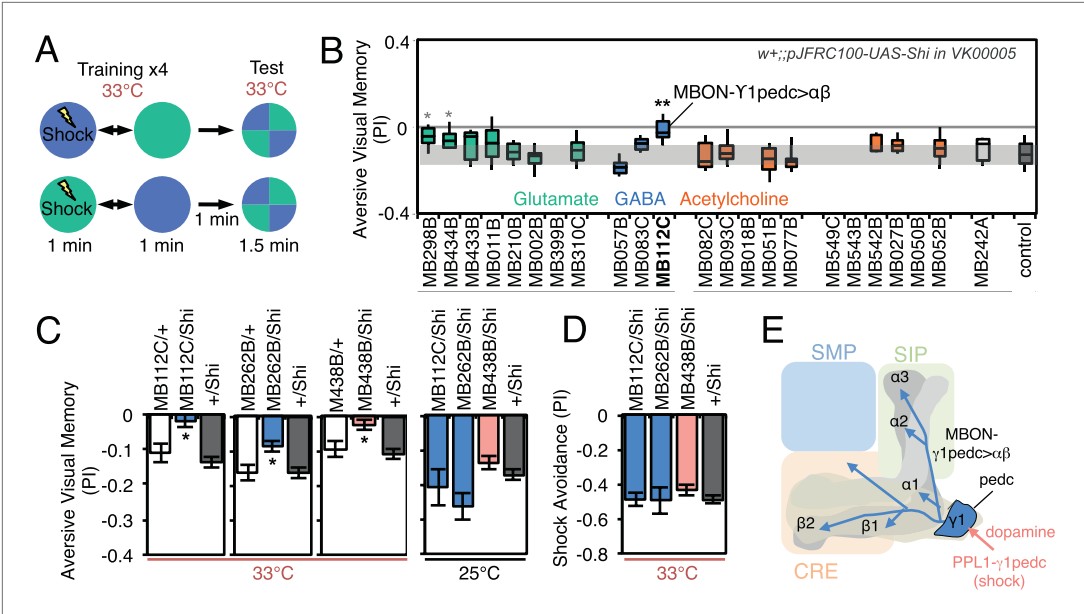

**Figure 8**. Requirement of MBONs for aversive visual memory. (**A**) Diagram of the training protocol used for aversive visual conditioning. Groups of 30–40 flies were trained in a circular arena with green or blue LED light and electric shock in a reciprocal manner, as in olfactory conditioning. Training was repeated four times and memory was tested immediately after the last training session. The PI corresponds to the mean of the [(number of flies in the blue quadrant minus number of flies in green quadrant)/total number of flies when blue was paired with electric shock] and [(number of flies in the green quadrant minus number of flies in the blue tube)/total number of flies when green was paired with electric shock]; See 'Materials and methods' for more detail. (**B**) Results for the requirement of MBONs in aversive visual conditioning. *pJFRC100-20XUAS-TTS-Shibire-ts1-p10* (*VK00005*) was used to block synaptic transmission. Flies were trained and tested at the restrictive temperature. MBONs have been grouped by neurotransmitter and color-coded. The bottom and top of each box represents the first and third quartile, and the horizontal line dividing the box is the median. The whiskers represent the 10th and 90th percentiles. The gray rectangle spanning the horizontal axis indicates the interquartile range of the control. Statistical tests are described in methods: *, $p < 0.05$; **, $p < 0.01$. The two lines marked with gray asterisks failed one of the control assays: MB434B/Shi did not differ significantly from MB434B/+ and MB298B had altered color preference. One line, MB112C, with significantly impaired memory passed all control assays (see panel **C**). (**C**) Blocking MBON-γ1pedc>α/β with either of two split-GAL4 lines (MB262B and MB112C) caused significant visual memory impairment compared to GAL4/+ and +/Shi controls. Blocking the output of PPL1-γ1pedc dopaminergic neurons (MB438B) also resulted in a strong defect. Split-GAL4/Shi flies were not impaired at the permissive temperature compared to +/Shi. Bars and error bars show mean and standard error of the mean. (**D**) Electric shock avoidance in untrained flies was not impaired at restrictive temperature compared to +/Shi. Bars and error bars show mean and standard error of the mean. (**E**) Diagram of the MB circuit for aversive visual memory. The GABAergic MBON-γ1pedc>α/β is required for conditioned color preference. The dopaminergic PPL1- γ1pedc neurons terminate in the same compartments where the dendrites of MBON-γ1pedc>α/β are found; they likely convey the electric shock punishment signal.

The following figure supplement is available for figure 8:

**Figure supplement 1**. Expression patterns of split-GAL4s that caused aversive visual memory phenotypes.

*Donlea et al., 2011*). Little is known about how the MB's sleep regulating functions are executed. As a first step in elucidating these mechanisms, we asked if activation of specific MBONs changed sleep pattern and amount.

Sleep was measured before, during, and after heat-gated activation of MBONs (*Figure 12A*) by expression of the temperature-gated dTrpA1 channel. We identified five MBON drivers that suppressed sleep and seven MBON drivers that increased sleep (*Figure 12B*). All five sleep-suppressing drivers express in glutamatergic MBONs (MBON-γ5β'2a, MBON-β'2mp, MBON-β'2a_bilateral and MBON-γ4>γ1γ2), while sleep-promoting drivers express either in GABAergic (MBON-γ3β'1 and

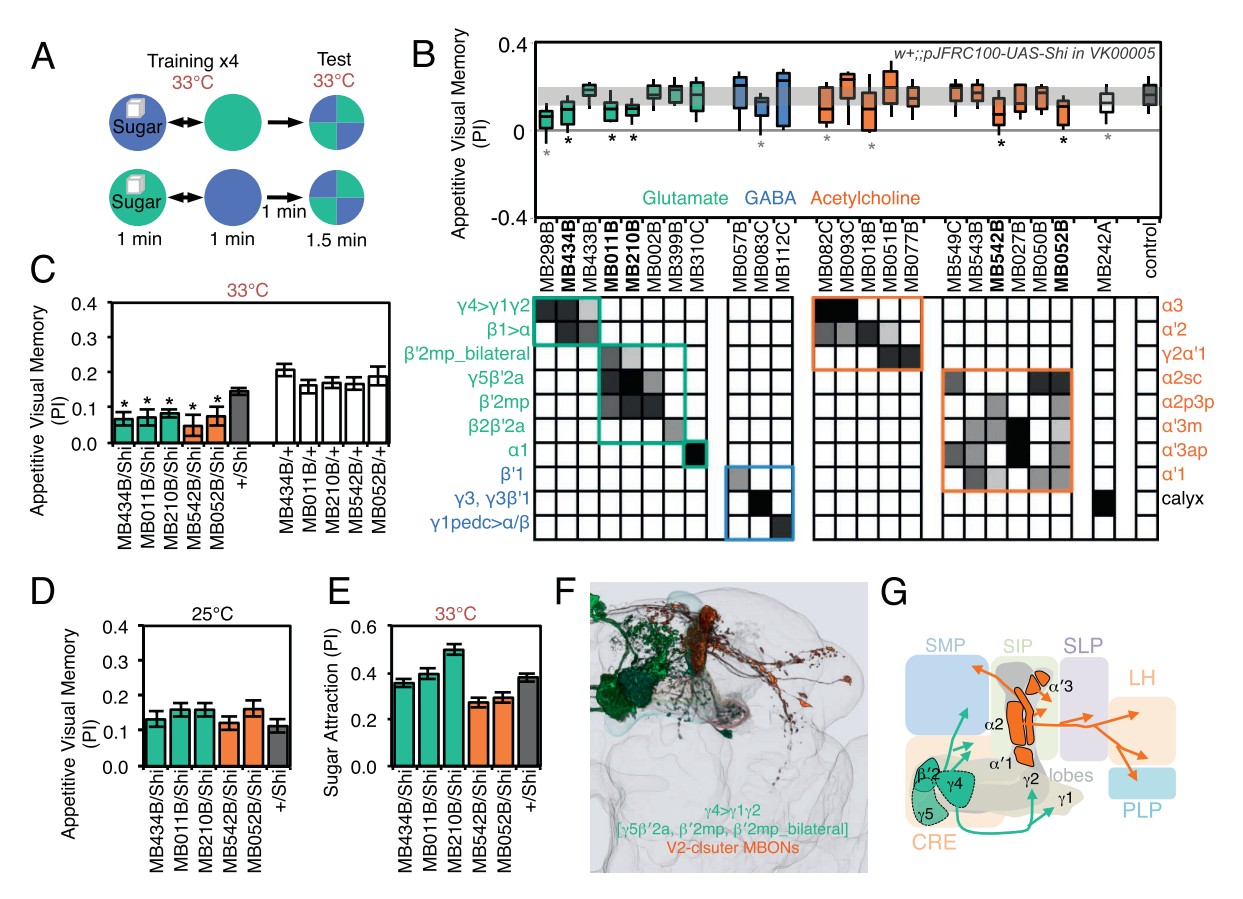

**Figure 9**. Requirement of MBONs for appetitive visual memory. (**A**) Diagram of the training protocol used for appetitive visual conditioning. Starved flies were trained with sugar and the PI calculated in the same manner as in aversive visual conditioning (see **Figure 8A**). Flies were trained and tested at the restrictive temperature. (**B**) Screening for MBONs required for aversive visual conditioning. *pJFRC100-20XUAS-TTS-Shibire-ts1-p10* (*VK00005*) was used to block synaptic transmission. MBONs have been grouped by neurotransmitter and color-coded. The bottom and top of each box represents the first and third quartile, and the line inside the box is the median. The whiskers represent the 10th and 90th percentiles. The gray rectangle spanning the horizontal axis indicates the interquartile range of the control. Statistical tests are described in methods: *, p < 0.05. Gray asterisks indicate lines that were not significantly different from GAL4/+ controls (MB083C, MB082C, MB018B and MB242A) or had altered color preference (MB298B). The relative expression levels produced by the split-GAL4 driver lines in each cell type (indicated in a gray scale) are based on imaging with *pJFRC2-10XUAS-IVS-mCD8::GFP* (*VK00005*). Images of the expression patterns of selected drivers are shown in **Figure 9—figure supplement 1**. (**C**) Visual appetitive memory was impaired at restrictive temperature by blocking the subsets of glutamate or acetylcholine MBONs labeled by MB434B, MB011B, MB210B, MB542B and MB052B compared to +/Shi and GAL4/+ controls. (**D**) No appetitive visual memory impairment was found at the permissive temperature compared to +/Shi. (**E**) Sugar attraction in untrained flies was not impaired at the restrictive temperature compared to +/Shi. (**F**) Rendering of MBONs implicated in appetitive visual memory. MBONs grouped by square brackets represent cases where the available set of driver lines do not allow assigning an effect to a single cell type, but only to the bracketed set. (**G**) Circuit diagram for appetitive visual memory. Glutamatergic MBONs (MBON-β'2mp and MBON-γ5β'2a) mediate visual appetitive memory. PAM neurons innervating these compartments of the MB lobes (see **Figure 1A**) are thought to convey the sucrose reward signal (**Vogt et al., 2014**). MBON-γ4>γ1γ2 feeds forward onto the dendrites of MBON-γ1pedc>α/β, a neuron required for aversive odor and aversive visual memory. Cholinergic V2 cluster neurons also appear to play a role.

The following figure supplement is available for figure 9:

**Figure supplement 1**. Expression patterns of split-GAL4s that caused appetitive visual memory phenotypes.

MBON-γ3) or cholinergic MBONs (MBON-γ2α'1); the neurotransmitter for MBON-calyx has not been determined (**Figure 12B**). Despite the 48 hr period of dTrpA1 activation, the effects of activation were reversible and the temperature shift had only minor effects on sleep in genetic control groups (**Figure 12C**). We retested the split-GAL4 drivers that showed a significant effect in the primary screening using a different dTrpA1 effector inserted at another genomic location, allowing

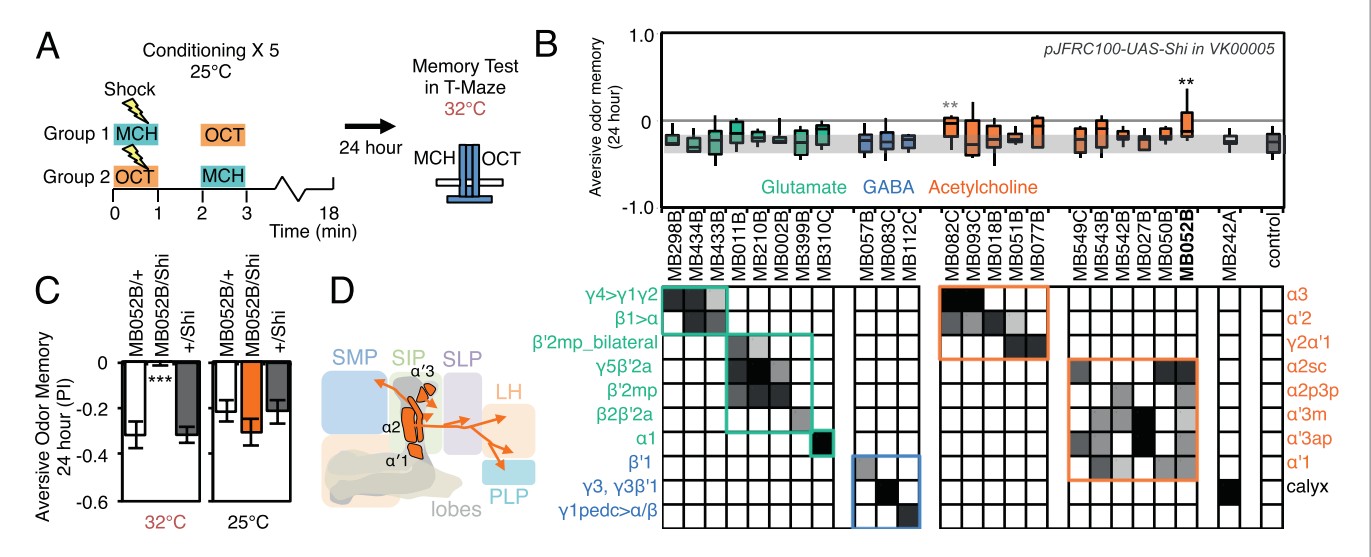

**Figure 10**. Requirement of long-term aversive odor memory. (**A**) Diagram of the conditioning protocol. Training sessions with electric shocks were repeated five times with 15 min intervals between training, all at the permissive temperature (25°C). Following training, flies were kept for 24 hr at 18°C, then shifted to the restrictive temperature (32°C) 30 min prior to test. The PI corresponds to the mean of the [(number of flies in the OCT tube minus number of flies in the MCH tube)/total number of flies when OCT was paired with electric shock] and [(number of flies in the MCH tube minus number of flies in the OCT tube)/total number of flies when MCH was paired with electric shock]. (**B**) Results of the primary screening for MBONs required for long-term aversive memory retrieval. *pJFRC100-20XUAS-TTS-Shibire-ts1-p10* (*VK00005*) was used to block synaptic transmission. The bottom and top of each box represents the first and third quartile, and the line inside the box is the median. The bottom and top of each box represents the first and third quartile, and the horizontal line dividing the box is the median. The whiskers represent the 10th and 90th percentiles. The gray rectangle spanning the horizontal axis indicates the interquartile range of the control. Statistical tests are described in methods: **: p < 0.01. The relative expression levels produced by the split-GAL4 driver lines in each cell type (indicated by the gray scale) are based on imaging with *pJFRC2-10XUAS-IVS-mCD8::GFP* (*VK00005*). Two lines, MB052B and MB082C showed significant memory impairment. MB082C drives expression in MBON-α3 and to a lesser extent in MBON-α'2. In subsequent control experiments, MB082C/Shi[ts] showed significant impairment compared to MB082C/+ and +/Shi[ts] (data not shown). However, we were unable to attribute this aversive LTM phenotype to MBON-α3, because blocking this cell type by other drivers did not result in a consistent phenotype (MB093C, *Figure 10B*; G0239, (*Placais et al., 2013*)). The requirement for MBON-α3 for aversive odor LTM has been reported, but it is unclear whether these neurons are required for canonical LTM after spaced training (*Pai et al., 2013*) or instead for so-called 'fasting LTM', a memory that mildly fasted flies can form after a single cycle of conditioning (*Hirano and Saitoe, 2013*; *Placais and Preat, 2013*; *Placais et al., 2013*). We note that these same two split-GAL4 driver lines for MBON-α3 (MB082C and MB093C) also showed discordant results in the ethanol memory assay (see below), suggesting some underlying difference between the two lines, perhaps in off-targeted expressions or genetic background. (**C**) MB052B/Shi showed memory impairment compared to MB052B/+ and +/Shi at the restrictive temperature but not at the permissive temperature. Statistical tests are described in methods: ***, p < 0.001. (**D**) Diagram of the output neurons that are required for long-term aversive odor memory. MB052B labels five cell types of cholinergic MBONs from the vertical lobes, four of them project to the lateral horn. Lines labeling only a subset of these MBONs gave only small effects when blocked.

us to assess expression more accurately since we had access to a reporter construct inserted at that site (*Figure 12—figure supplement 1*); these assays confirmed the initial results for all but one of the split-GAL4 driver lines (*Figure 12D*). None of the lines showed significant effects on general locomotion as assessed by video tracking (see 'Materials and methods'). We identified distinct MBONs that either decrease (*Figure 12E*) or increase sleep (*Figure 12F*). The subset of MBONs that promoted sleep was similar to the subset in which CsChrimson activation was attractive; conversely, the MBON subset that promoted wakefulness was similar to the subset in which CsChrimson activation was aversive.

The projection patterns of MBONs provide insight into how these bidirectional signals might be integrated in the fly brain (*Figure 12G*). For example, the axons of the sleep promoting cholinergic MBON-γ2α'1 and wake promoting glutamatergic MBON-β'2mp project their termini to the same location in the brain (*Video 6*). These circuit arrangements of sleep- and wake-promoting neurons may facilitate the transition between the sleep and wake behavioral states by providing opposing inputs to shared downstream targets.

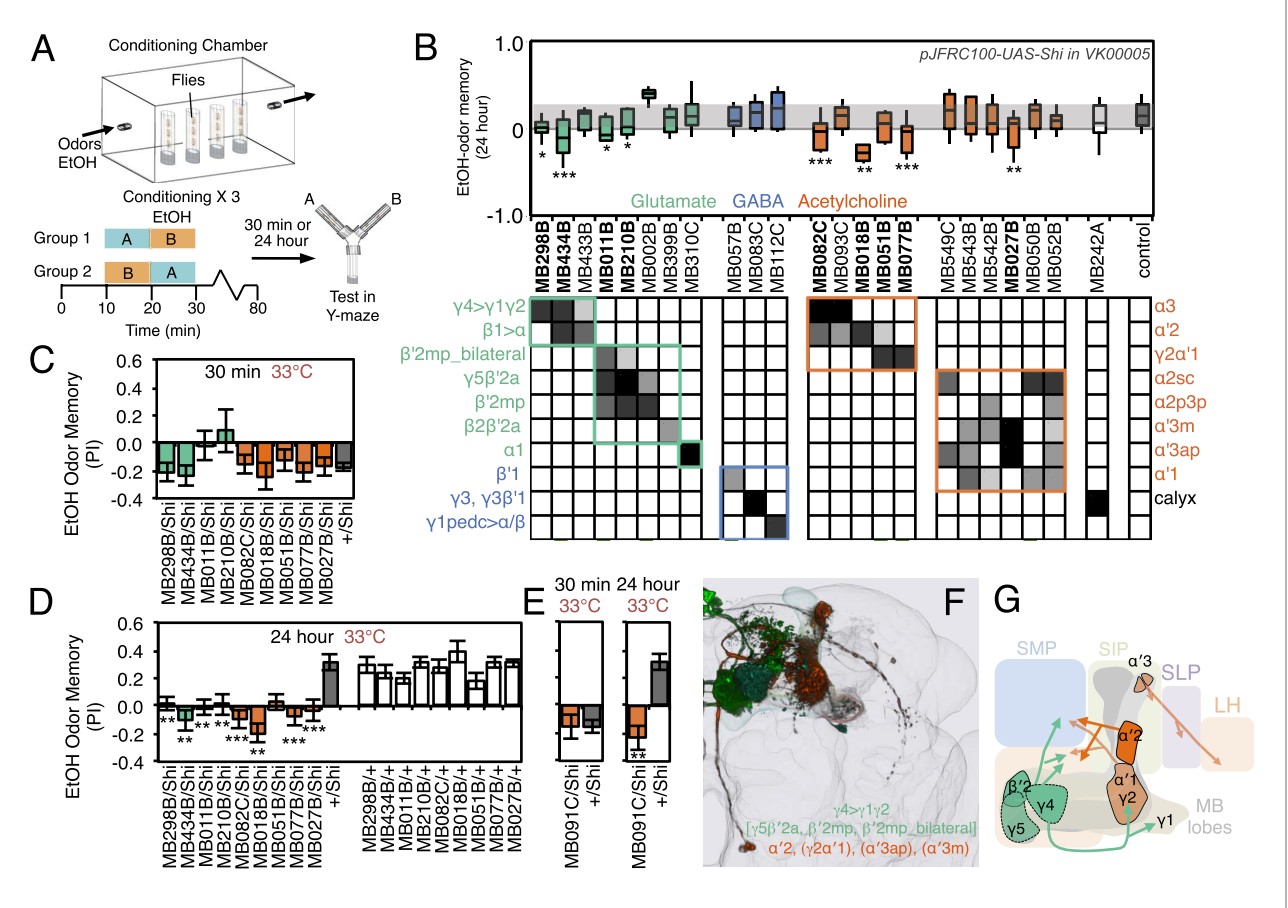

**Figure 11**. Requirement of MBONs for appetitive odor-ethanol intoxication memory. (**A**) Diagram of odor conditioning using ethanol vapor. Conditioned ethanol preference is measured by presenting two odors in sequence, one in the presence of an intoxicating dose of ethanol, three times for 10 min with 50-min breaks between exposures, followed by testing for preference between the two odors in a Y-maze in the absence of ethanol either 30 min or 24 hr after training. Wild-type flies find the ethanol-paired odor to be aversive when tested 30 min after training, but appetitive 24 hr after training. (**B**) Results of assays for the requirement of MBONs in ethanol-induced appetitive odor memory. *pJFRC100-20XUAS-TTS-Shibire-ts1-p10* (*VK00005*) was used to block synaptic transmission during training and test; flies were kept at the permissive temperature for the 24 hr between training and test. MBONs have been grouped by neurotransmitter and color-coded. The bottom and top of each box represents the first and third quartile, and the horizontal line dividing the box is the median. The whiskers represent the 10th and 90th percentiles. The gray rectangle spanning the horizontal axis indicates the interquartile range of the control. Statistical tests are described in methods: *, p < 0.05; **, p < 0.01; ***, p < 0.001. The relative expression levels produced by the split-GAL4 driver lines in each cell type (indicated by the gray scale) are based on imaging with *pJFRC2-10XUAS-IVS-mCD8::GFP* (*VK00005*). Images of the expression patterns of selected drivers are shown in *Figure 11—figure supplement 1*. (**C**) 30-min ethanol-induced odor memory. (**D**) 24-hr ethanol-induced appetitive odor memory compared to +/Shi and GAL4/+ genetic controls. (**E**) Blocking transmission in MBON-α'2 with MB091C resulted in a failure to switch from aversive to appetitive memory for ethanol intoxication, confirming the result seen with MB018B. (**F**) Renderings of the MBONs implicated in 24 hr ethanol odor memory. MBONs grouped by square brackets represent cases where the available set of driver lines do not allow assigning an effect to a single cell type, but only to that set of MBONs. MBONs cell types in parentheses indicate cases where the data implicating them are only suggestive. (**G**) Schematic of potential circuits for appetitive ethanol memory. Cholinergic outputs from the MBON-α'2 and MBON-γ2α'1, and glutamatergic MBON-γ4>γ1γ2, MBON-γ5β'2a and MBON-β'2mp are implicated in appetitive ethanol memory. MBONs for which the data shows inconsistency across driver lines (MBON-α3 and MBON-γ2α'1) are shown in a lighter color; additional experiments will be required to resolve the role of these MBONs.

The following figure supplement is available for figure 11:

**Figure supplement 1**. Expression patterns of split-GAL4s that caused odor-ethanol intoxication memory phenotypes.

## Discussion

In the insect brain, a sparse and non-stereotyped ensemble of Kenyon cells represents environmental cues such as odors. The behavioral response to these cues can be neutral, repulsive or attractive, influenced by the prior experiences of that individual and how dopaminergic and other modulatory inputs

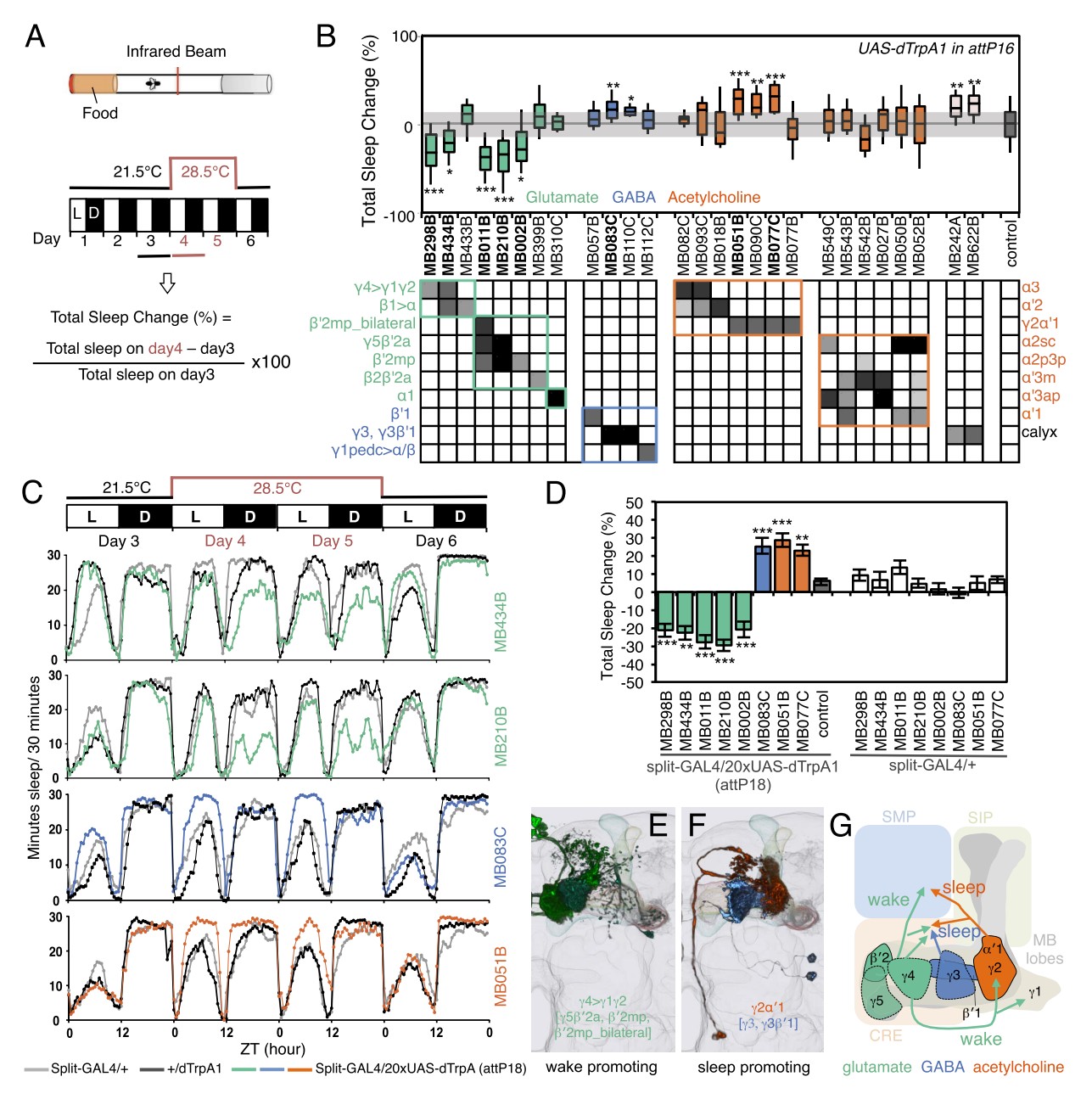

**Figure 12**. MBONs bi-directionally regulate sleep. (**A**) Top: Schematic of the experimental apparatus used for assaying sleep. Single flies are placed in a tube and activity is measured by counting the number of times the fly crosses an infrared beam. Bottom: Diagram of the experimental assay. Sleep was measured at 21.5°C for 3 days to establish the baseline sleep profile. Flies were then shifted to 28.5°C for 2 days to increase activity of the targeted cells by activating the dTrpA1 channel, and then returned to 21.5°C after activation to assay recovery. The effect of MBON activation on sleep amount is quantified as percentage change in sleep. Negative and positive values indicate decreased and increased sleep, respectively. (**B**) Box plots of change in sleep induced by dTrpA1 [*UAS-dTrpA1* (*attP16*); (***Hamada et al., 2008***)] mediated activation of the neurons targeted by each of the indicated split-GAL4 driver lines. MBONs are grouped by neurotransmitter and color-coded as indicated. The bottom and top of each box represents the first and third quartile, and the horizontal line dividing the box is the median. The whiskers represent the 10th and 90th percentiles. The gray rectangle spanning the horizontal axis indicates the interquartile range of the control. Statistical tests are described in methods: *, p < 0.05; **, p < 0.01; ***, p < 0.001. The matrix below the box plots shows the relative levels of expression (indicated by the gray scale) in the cell types observed with each driver line using *pJFRC200-10XUAS-IVS-myr::smGFP-HA* (*attP18*). Images of the expression patterns of selected drivers are shown in ***Figure 12—figure supplement 1***. (**C**) Sleep profiles of four split-GAL4 lines are shown. Each plot shows a 4-day period starting with subjective dawn. Sleep duration (min/30 min) on day 3 (21.5°C; permissive temperature), days 4, 5 (28.5°C; non-permissive temperature) and day 6 (21.5°C; permissive temperature) are plotted (colored line,

*Figure 12. Continued on next page*

*Figure 12. Continued*

split-GAL4 line in combination with *pJFRC124-20XUAS-IVS-dTrpA1* (*attP18*); black line, represents +/dTrpA1; gray line, split-GAL4/+). (**D**) Sleep phenotypes were replicated with *pJFRC124-20XUAS-IVS-dTrpA1* (*attP18*) for the eight drivers shown, but not for MB242A (not shown). Corresponding split-GAL4/+ flies showed normal sleep. (**E**, **F**) Renderings of MBONs responsible for the decreasing (**E**) or increasing (**F**) sleep. MBONs are color-coded based on their putative transmitter. (**G**) Diagram of MBONs responsible for the sleep regulation. MBONs are color-coded based on their putative transmitter as indicated. The wake promoting glutamatergic MBON-γ5β′2a, MBON-β′2mp and MBON-β′2mp_bilateral converge with the sleep promoting cholinergic MBON-γ2α′1 and GABAergic MBON-γ3 and MBON-γ3β′1 in the SMP and CRE. The wake-promoting glutamatergic MBON-γ4>γ1γ2 terminates in the dendritic region of MBON-γ2α′1.

The following figure supplement is available for figure 12:

**Figure supplement 1**. Expression patterns of split-GAL4s that caused sleep phenotypes.

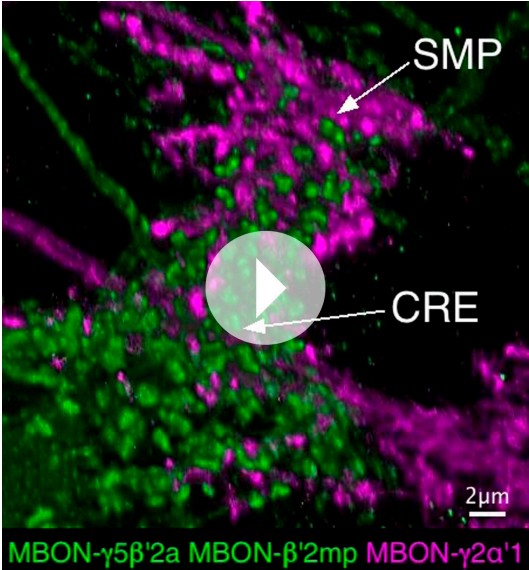

**Video 6**. Convergence of glutamatergic and cholinergic MBONs. Rendering of two-color labeling for the glutamatergic MBON-β′2mp and weakly labeled MBON-γ5β′2a (MB074C; green) and cholinergic MBON-γ2α′1 (*R25D01-LexAp65* in attP40; magenta). A small area dorsal to the MB medial lobes (an area that includes part of the CRE and the SMP) is shown.

have changed the weight of its KC-MBON synapses. MBONs are thought to encode the predictive value associated with a stimulus. A fly would then use that information to bias its selection of behavioral responses in an ever-changing environment. In the accompanying paper (*Aso et al., 2014*), we describe in detail the projection patterns of the MBON and DAN cell types that comprise the MB lobes in *Drosophila*. In this report, we begin the process of determining the nature of the information conveyed by MBONs. We also correlate specific MBON cell types with roles in associative memory and sleep regulation (*Figure 13*). Our anatomical and behavioral results lay the groundwork for understanding the circuit principles for a system of memory-based valuation and action selection.

## The ensemble of MBONs represents valence and biases choice

We demonstrated that optogenetic activation of MBONs in untrained flies can induce approach or avoidance. The ability of the MBONs to induce changes in behavior in the absence of odors suggests that MBONs can bias behavior directly. This observation is consistent with a recent study showing that flies are able to associate artificial activation of a random set of KCs—instead of an odor stimulus—with electric shock, and avoid reactivation of the same set of KCs in the absence of odors (*Vasmer et al., 2014*), a result that recapitulates a finding in the potentially analogous piriform cortex of rodents (*Choi et al., 2011*). We found that the sign of the response to MBON activation was highly correlated with neurotransmitter type; all the MBONs whose activation resulted in avoidance were glutamatergic, whereas all the attractive MBONs were cholinergic or GABAergic.

By tracking flies as they encounter a border between darkness and CsChrimson-activating light, we showed that activation of an MBON can bias walking direction. Although activation of glutamatergic MBONs repelled flies, the avoidance behaviors were not stereotyped; flies showed a variety of motor patterns when avoiding the red light. This observation implies that MBONs are unlikely to function as command neurons to drive a specific motor pattern, as has been observed, for example, in recently identified descending neurons that induce only stereotyped backward walking (*Bath et al., 2014*; *Bidaye et al., 2014*).

Rather, we view fly locomotion as a goal-directed system that uses changes in MBON activity as an internal guide for taxis. For example, walking in a direction that increases the relative activity of

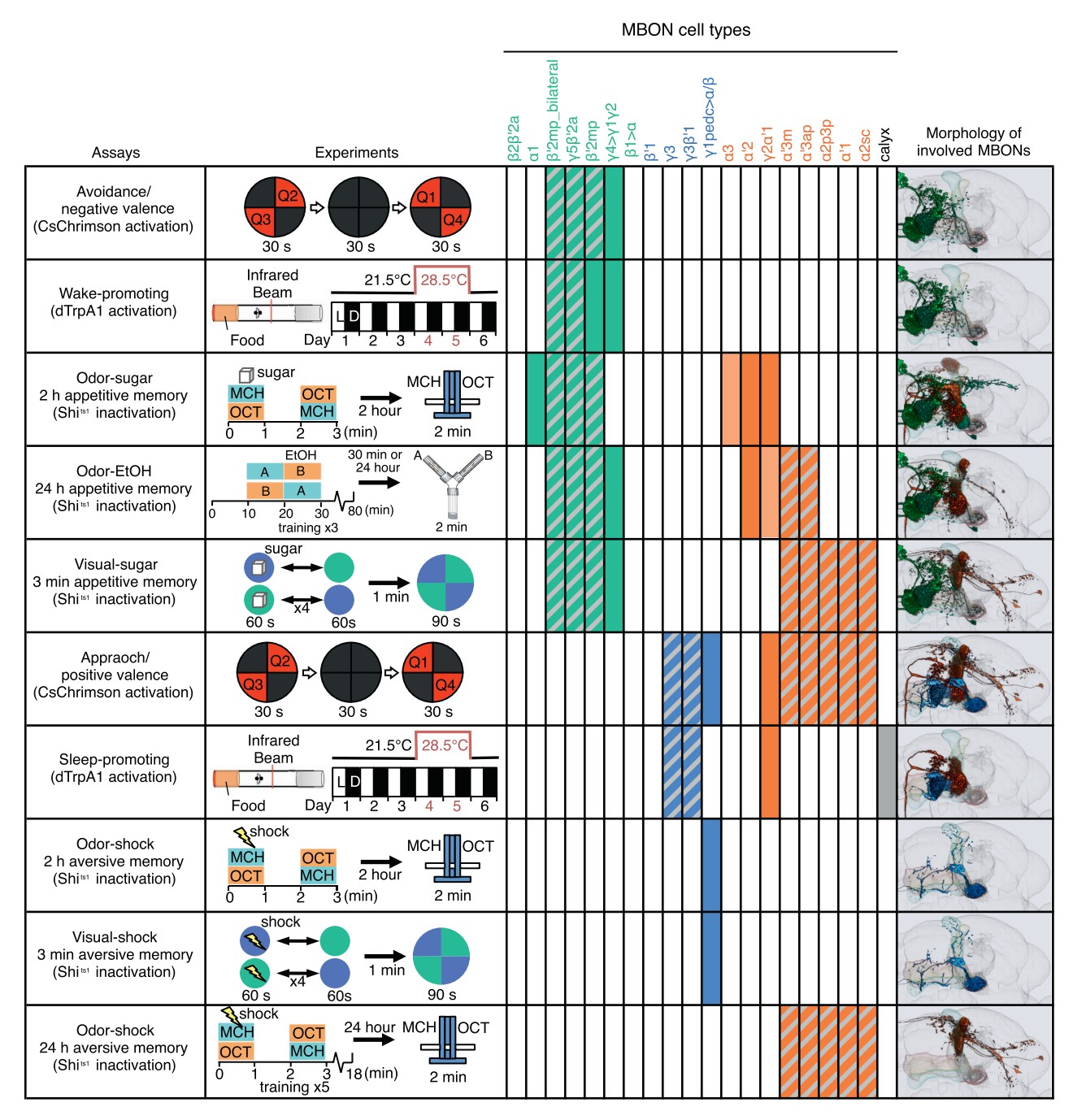

**Figure 13.** A map of MBON functions. A matrix summarizing the MBON cell types identified in each of the behavior assays is shown. In the cells of the matrix representing MBON cell types, solid colors were assigned when manipulation of that cell type alone caused a significant behavioral phenotype. Hatched cells indicate cases where we observed a significant behavioral phenotype, but the driver line used labeled a combination of cell types. In these cases, our data only allow us to assign the phenotypic effect to that group of MBONs. The light colored cells indicate two cases where our data suggested the involvement of a cell type, but was not conclusive. See Results for details. Avoidance, attraction, wake-promoting and sleep-promoting phenotypes reflect the results of activating MBONs; the other assays measured the effects of blocking synaptic transmission. Activation of glutamatergic MBONs repelled flies and promoted wakefulness, whereas activation of GABAergic or cholinergic MBONs attracted flies and promoted sleep. In general, more cell types were involved in appetitive memory than in aversive memory. Partially overlapping set of cholinergic and glutamatergic MBONs were required for all three appetitive memory assays. The same GABAergic MBON-γ1pedc>α/β was required for short-term olfactory and visual memory in an assay protocol that did not distinguish between formation or retrieval, whereas long-term aversive odor memory retrieval required the V2 cluster MBONs.

aversive-encoding MBONs, which would occur as a fly approaches an odorant it had previously learned to associate with punishment (or when a fly expressing CsChrimson in an avoidance-inducing MBON approaches the CsChrimson activating light as in *Figure 5E*), signals the locomotive system to turn around and walk the other direction. Detailed studies of locomotor circuitry will be required to determine the mechanisms of executing such taxic behaviors and should help elucidate how MBON inputs guide this system.

In this view, the MBON population functions as neither a purely motor nor a purely sensory signal. From the motor perspective, as described above, MBONs bias locomotive outcomes rather than dictate a stereotyped low-level motor program. From the sensory perspective, we have shown that the same MBONs can be required for experience-dependent behavioral plasticity irrespective of whether a conditioned stimulus is a color or an odor, and irrespective of the specific identity of the odor. Taken together with the fact that MBONs lie immediately downstream of the sites of memory formation, these observations support our proposal that MBONs convey that a stimulus has a particular value— but not the identity of the stimulus itself. This contrasts with sensory neurons whose activity can also induce approach or avoidance, but which do convey the stimulus per se. In mammals, neural representations of abstract variables such as 'value', 'risk' and 'confidence' are thought to participate in cognition leading to action selection (for example, see *Kepecs et al., 2008*; *Kiani and Shadlen, 2009*; *Levy et al., 2010*; *Yanike and Ferrera, 2014*). From the point of view of this framework, the MBON population representing the value of a learned stimulus and informing locomotion might be operationally viewed as a cognitive primitive.

Co-activating multiple MBON cell types revealed that the effects of activating different MBONs appear to be additive; that is, activating MBONs with the same sign of action increases the strength of the behavioral response, whereas activating MBONs of opposite sign reduces the behavioral response. Thus, groups of MBONs, rather than individual MBONs, likely act collectively to bias behavioral responses. Consistent with the idea of a distributed MBON population code, all 19 MBON cell types imaged so far show a calcium response to any given odor (Hige et al., unpublished, *Sejourne et al., 2011*; *Pai et al., 2013*; *Placais et al., 2013*).

If it is the ensemble activity of a large number of MBONs that determines memory-guided behavior, how can local modulation of only one or a few MB compartments by dopamine lead to a strong behavioral response? Activation of a single DAN such as PPL1-γ1pedc that innervates a highly localized region of the MB can induce robust aversive memory, yet the odor associated with the punishment will activate MBONs from all compartments, including MBONs that can drive approach as well as those that drive avoidance. We propose that, in response to a novel odor stimulus, the activities of MBONs representing opposing valences may initially be 'balanced', so that they do not impose a significant bias. Behavior would then be governed simply by any innate preference a fly might have to that odor, using neuronal circuits not involving the MB. Now suppose an outcome associated with that stimulus is learned. Such learning involves compartment-specific, dopamine-dependent plasticity of the KC-MBON synapses activated by that stimulus. If that occurs, the subsequent ensemble response of the MBONs to that stimulus would no longer be in balance and an attraction to, or avoidance of, that stimulus would result. Consistent with this idea, eliminating MB function by disrupting KCs, which are nearly 10% of neurons in the central brain, had surprisingly minor effects on odor preference (*de Belle and Heisenberg, 1994*, *Heisenberg et al., 1985*; *McGuire et al., 2001*). *Figure 14* shows a conceptual model of how this could be implemented at the level of neuronal circuits.

## Synaptic plasticity underlying associative memory

Recent studies of dopamine signaling have implicated distinct sites of memory formation within the MB lobes (summarized in *Figure 1A*) (*Schroll et al., 2006*; *Claridge-Chang et al., 2009*; *Aso et al., 2010*, *2012*; *Burke et al., 2012*; *Liu et al., 2012*; *Perisse et al., 2013*). Consistent with this large body of work, we found that one type of PPL1 cluster DAN, PPL1-γ1pedc, played a central role in formation of aversive memory in both olfactory (*Figure 6*; see also *Aso et al., 2012*; *Aso et al., 2010*) and visual learning (*Figure 8*) paradigms. This DAN also mediates aversive reinforcement of bitter taste (*Das et al., 2014*). For appetitive memory, PAM cluster DANs that innervate other regions of the MB lobes, in particular the compartments of glutamatergic MBONs, are sufficient to induce appetitive memory (*Yamagata et al., in press*) (*Burke et al., 2012*; *Liu et al., 2012*) (*Perisse et al., 2013*). These results strongly suggest that the synaptic plasticity underlying appetitive and aversive memory generally occurs in different compartments of the MB lobes.

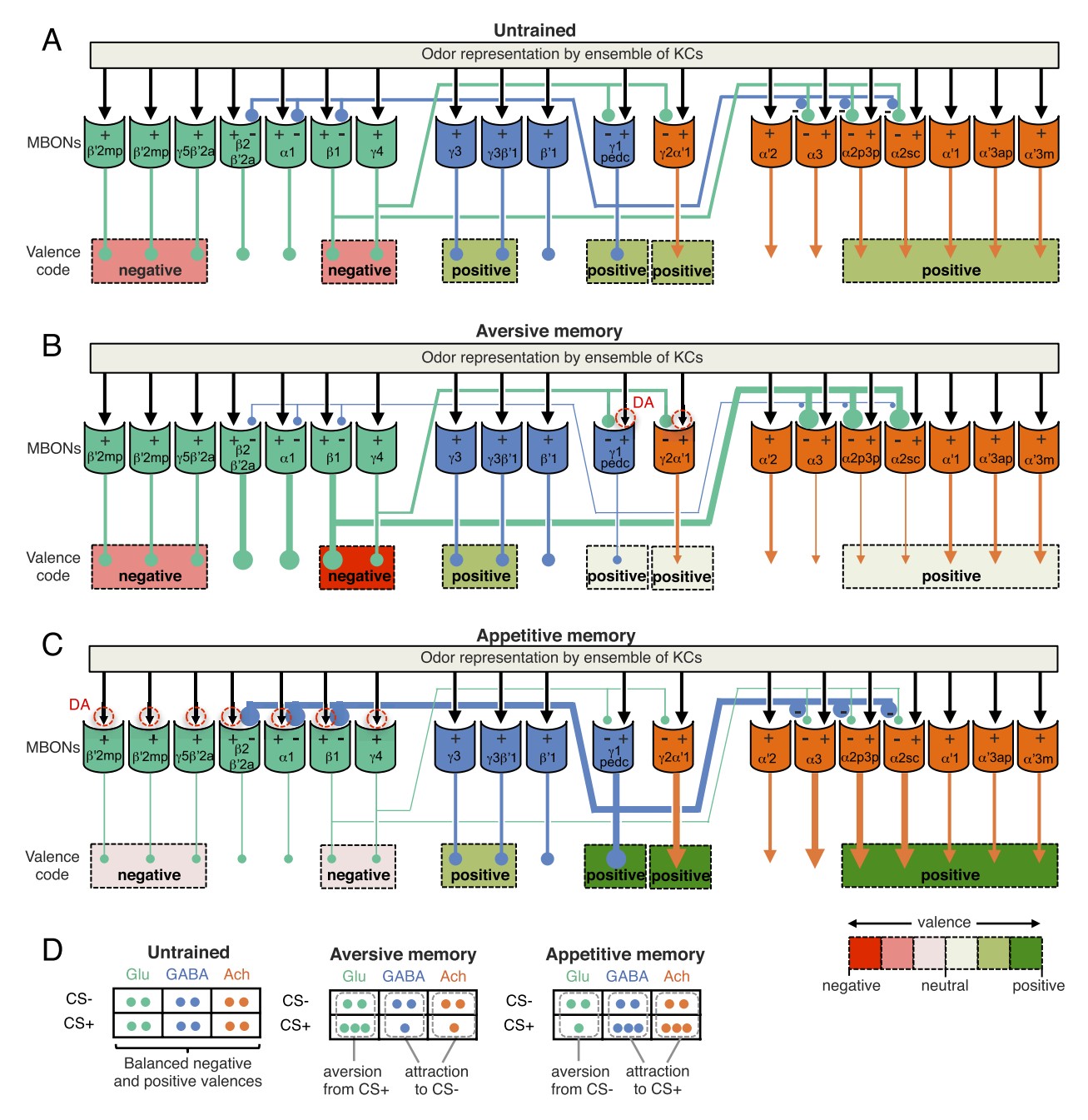

**Figure 14**. A simplified circuit model for the encoding of learned valence by MBONs ensemble and feedforward network. (**A**) In response to an odor, a sparse ensemble of Kenyon cells provides excitatory synaptic input to MBONs (black arrows; Hige et al., unpublished) (***Cassenaer and Laurent, 2007***). Glutamatergic (Glu; green), GABAergic (GABA; blue) and cholinergic (Ach; orange) MBONs all receive KC input; the names of MBONs are based on the lobe compartment where their dendrites arborize. CsChrimson activation of some glutamatergic MBONs can be repulsive, whereas activation of GABA or cholinergic MBONs can be attractive (color coded as indicated by the scale at bottom right) (***Figure 2C***). We often only observed significant behavioral effect with combinations of cell types (indicated by dashed lines grouping multiple cell types). MBON-γ1pedc>α/β, and glutamatergic MBONs, MBON-β1>α and MBON-γ4>γ1γ2, have synaptic terminals inside the compartments of MB lobes (***Aso et al., 2014***). While the microcircuits within each MB compartment remain to be elucidated, our light level anatomical studies have enumerated the cell types present in each compartment. The α2 and α3 compartments contain the dendrites of cholinergic MBONs and are targeted by both the GABAergic MBON-γ1pedc>α/β and the glutamatergic MBON-β1>α. These MBONs cover only a small fraction of volume in their target compartments (see ***Figure 6—figure supplement 1***); since they could contact only a fraction of Kenyon cells, we propose that they target MBONs directly. Here we hypothesize that glutamate is inhibitory to cholinergic and GABAergic MBONs, and GABA is inhibitory to glutamatergic MBONs (inhibitory connections are indicated by circles). The thickness of lines and size of their endings
*Figure 14. Continued on next page*

*Figure 14. Continued*

are meant to indicate activity levels. (**B**) PPL1- γ1pedc DANs play a major role to mediate punishment signals to the MB for formation of aversive memory together with minor contribution from other DANs including PPL1-γ2α'1 (*Aso et al., 2012*). If an aversive memory is formed by the simultaneous presentation of an odor and punishment and results in a synaptic depression of KC terminals by dopamine (represented by red dashed circles), the response of the GABAergic MBONs to the CS+ would be depressed (see text). Reduced GABAergic inhibitory input would then increase the CS+ response of glutamatergic MBONs, whereas the CS+ response of cholinergic MBONs would be reduced because of dis-inhibition of the inhibitory glutamatergic MBONs. The end result is enhanced activity of aversion-mediating glutamatergic MBONs together with the reduced activity of attraction-mediating GABAergic and cholinergic MBONs in response to CS+. The CS- is represented by a different set of KCs whose synaptic connections to the MBONs would not be expected to be modified by training and so the responses of the ensemble of MBONs to the CS- would remain balanced. The change in response of the MBONs to the CS+, relative to their unchanged response to the CS-, biases choice toward the CS+ (see diagram in panel **D**). This model is consistent with the essential role of MBON-γ1pedc>α/β in aversive memory. Also, cholinergic MBONs in V2 cluster have been shown to reduce their response to an odor after olfactory conditioning with electric shock (Séjourné et al., Nat. Neurosci., 2011), although we detected their requirement only for long-term memory, but not for 2 hr memory. This model predicts a role for glutamatergic MBONs, but we did not observe significant effect by blocking subsets of glutamtergic MBONs (*Figures 6 and 8*). Thus, to test this model, it will be necessary to block broader sets of glutamatergic MBON cell types by using combinations of split-GAL4 drivers. (**C**) In contrast to aversive memory in which one type of DAN (PPL1-γ1pedc) plays a major role in memory formation, reward signals are mediated by a distributed set of PAM cluster DANs that innervate the compartments of glutamatergic MBONs (*Yamagata et al., in press*) (*Burke et al., 2012*; *Liu et al., 2012*; *Perisse et al., 2013*). If an appetitive memory is formed by synaptic depression of KC terminals in response to dopamine release, the response of the glutamatergic MBONs to the CS + would be depressed. The resultant reduced glutamatergic inhibitory input to the GABAergic and cholinergic MBONs would increase their response to the CS+. In turn, increased GABAergic input to glutamatergic MBONs may further amplify and stabilize the initial effects of plasticity. The end result would be reduced activity of aversion-mediating glutamatergic MBONs together with the increased activity of attraction-mediating GABAergic and cholinergic MBONs in response to CS+. This model is consistent with requirement of glutamatergic MBONs: blocking the M4/M6 cluster MBONs (MBON-γ5β'2a, MBON-β'2mp and MBON-β'2mp_bilateral; *Table 1*) by MB011B resulted in memory impairment for all three appetitive memory assays (*Figures 7, 9 and 11*). Blocking the MBON-γ4>γ1γ2 and MBON-β1>α by MB434B resulted in memory impairment in two of three assays (*Figures 9 and 11*). While we did not detect a requirement for the GABAergic MBON-γ1ped>α/β in appetitive memory, previous study have shown that dopamine input to the γ1 and pedc suppresses expression of appetitive memory in fed flies (*Krashes et al., 2009*), indicating some role of MBON-γ1ped>α/β in appetitive memory. Blocking cholinergic MBONs in the V3/V4 cluster (MBON-γ2α'1, MBON-α'2 and MBON-α3) resulted in memory impairment in appetitive odor memories (*Figures 7 and 11*) (*Placais et al., 2013*) but not in appetitive visual memory (*Figure 9*). Some of driver lines for the cholinergic MBONs in the V2 cluster showed impairment of appetitive memory in all three assays (*Figures 7, 9 and 11*), although our data did not allow mapping to the resolution of specific cell types due to inconsistent results obtained using other lines. (**D**) In the matrix shown, the number of circles represents the activity levels of MBONs in response to the CS+ odor (the odor that is paired with the unconditioned stimulus during conditioning) and the CS− odor (a control odor). In untrained flies, the activities of MBONs for opposing effects are balanced. Dopamine modulation breaks this balance to bias the choice between CS+ and CS−.

The sign of preference we observed in response to CsChrimson activation of particular MBONs was, in general, opposite to that of the memory induced by dopaminergic input to the corresponding MB compartments. For example, activation of MBON-γ1pedc>α/β and MBON-γ2α'1 attracted flies (*Figure 2*), whereas DAN input to these regions induced aversive memory (*Figure 6G*) (*Aso et al., 2010*, *2012*). Conversely, activation of glutamatergic MBONs repelled flies (*Figure 2*), while DAN input to the corresponding regions is known to induce appetitive memory (*Yamagata et al., in press*) (*Burke et al., 2012*; *Liu et al., 2012*) (*Perisse et al., 2013*). These results are most easily explained if dopamine modulation led to synaptic depression of the outputs of the KCs representing the CS + stimulus. Consistent with this mechanism, the PE1 MBONs in honeybees (*Okada et al., 2007*) as well as the V2 cluster MBONs in *Drosophila* (*Sejourne et al., 2011*) reduce their response to a learned odor and depression of KC-MBON synapses has been shown for octopamine modulation in the locust MB (*Cassenaer and Laurent, 2012*). Moreover, long-term synaptic depression is known to occur in the granular cell synapses to Purkinje cells in the vertebrate cerebellum (*Ito et al., 1982*), a local neuronal circuit with many analogies to the MB (*Schurmann, 1974*; *Laurent, 2002*; *Farris, 2011*). Other mechanisms are also possible and multiple mechanisms are likely to be used. For example, dopamine may modulate terminals of KCs to potentiate release of an inhibitory cotransmitter such as short neuropeptide F, which has been demonstrated to be functional in KCs (*Knapek et al., 2013*) and hyperpolarizes cells expressing the sNPF receptor (*Shang et al., 2013*). RNA profiling of MBONs should provide insights into the molecular composition of synapses between KCs and MBONs. It is also noteworthy that the effect of dopamine can be dependent on the activity status of Kenyon cells; activation of PPL1-γ1pedc together with odor presentation induces memory, while its activation without an odor has been reported to erase memory (*Berry et al., 2012*; *Placais et al., 2012*). In the vertebrate basal ganglia, dopamine dependent synaptic plasticity important for aversive and appetitive learning is known to result in both synaptic potentiation and synaptic depression (*Shen et al., 2008*).

## Requirements for MBONs differ between behaviors

In this study, we looked at the effects of selectively and specifically manipulating the activities of a comprehensive set of MBONs on several behaviors. As a consequence, we gained some insights into the extent to which the relative importance of particular MBONs differed between behaviors (*Figure 13*). Most obvious was the segregation between appetitive and aversive behaviors. For example, we found that blocking MBON-γ1pedc>α/β impaired both short-term aversive odor and visual memory, suggesting a general role in aversive memory independent of modality. Conversely, a subset of glutamatergic MBONs was required in all three appetitive memory assays. It still remains to be demonstrated that the outputs of these MBONs are required transiently during memory retrieval. Nevertheless, CsChrimson activation experiments demonstrate that activation of these MBONs can directly and transiently induce attraction and avoidance behaviors.

In the cases described above, the DANs and MBONs mediating a particular behavior innervate the same regions of MB lobes. We also found cases where the DANs and MBONs required for a behavior do not innervate the same compartments of the MB lobes. For example, even though several cholinergic MBONs are required for appetitive memory (*Placais et al., 2013*), the compartments with cholinergic MBONs do not receive inputs from reward-mediating PAM cluster DANs, but instead from PPL1 cluster DANs that have been shown to be dispensable for odor-sugar memory (*Schwaerzel et al., 2003*) (*Figure 1A*). What accounts for this mismatch? Perhaps these cholinergic MBONs' primarily function is in memory consolidation rather than retrieval. But the fact that CsChrimson activation of the cholinergic MBON-γ2α′1 and V2 cluster MBONs resulted in attraction, strongly suggests that at least some of the cholinergic MBONs have a role in directly mediating the conditioned response. Indeed, previous studies found a requirement for cholinergic MBONs (the V2 cluster and MBON-α3) during memory retrieval (*Sejourne et al., 2011*; *Pai et al., 2013*; *Placais et al., 2013*). One attractive model is that requirement of cholinergic MBONs originates from the transfer of information between disparate regions of the MB lobes through the inter-compartmental MBONs connections within the lobes or by way of connections outside the MB, like those described in the next two sections.

## Roles for the multilayered network of MBONs

The multilayered arrangement of MBONs (see Figure 17 of the accompanying manuscript) (*Aso et al., 2014*) provides a circuit mechanism that enables local modulation in one compartment to affect the response of MBONs in other compartments. Once local modulation breaks the balance between MBONs, these inter-compartmental connections could amplify the differential level of activity of MBONs for opposing effects (*Figure 14*). For example, the avoidance-mediating MBON-γ4>γ1γ2 targets the compartments of attraction-mediating MBON-γ2α′1 and MBON-γ1pedc>α/β (*Figure 14*).

This network topology might also provide a fly with the ability to modify its sensory associations in response to a changing environment (see Discussion in the accompanying manuscript) (*Aso et al., 2014*). Consider the α lobe. Previous studies and our results indicate that circuits in the α lobe play key roles in long-term aversive and appetitive memory (*Figure 10*) (*Pascual and Preat, 2001*; *Isabel et al., 2004*; *Yu et al., 2006*; *Blum et al., 2009*; *Sejourne et al., 2011*; *Cervantes-Sandoval et al., 2013*; *Pai et al., 2013*; *Placais et al., 2013*). The α lobe is targeted by MBONs from other compartments and comprises the last layer in the layered output model of the MB (*Figure 1B*; see also Figure 17 of the accompanying manuscript) (*Aso et al., 2014*). The GABAergic MBON-γ1pedc>α/β and the glutamatergic MBON-β1>α both project to α2 and α3, where their axonal termini lie in close apposition (see Figure 17 of the accompanying manuscript) (*Aso et al., 2014*). DAN input to the compartments housing the dendrites of these feedforward MBONs induces aversive and appetitive memory, respectively (this work) (*Perisse et al., 2013*; *Yamagata et al., in press*). As pointed out in (*Aso et al., 2014*), this circuit structure is well-suited to deal with conflicts between long-lasting memory traces and the need to adapt to survive in a dynamic environment where the meaning of a given sensory input may change. To test the proposed role of the layered arrangement of MBONs in resolving conflicts between old memories and new sensory inputs, we will also need behavioral paradigms that, unlike the simple associative learning tasks used in our current study, assess the neuronal requirements for memory extinction and reversal learning.

## Network nodes for integrating innate and learned valences

The neuronal circuits that are downstream of the MBONs and that might read the ensemble of MBON activity remain to be discovered. However, the anatomy of the MBONs suggests that, at least in some

cases, summation and canceling effects may result from convergence of MBON terminals on common targets (*Aso et al., 2014*). For example, the terminals of the sleep-promoting cholinergic MBON-γ2α'1 overlap with terminals of wake-promoting glutamatergic MBONs (γ5β'2a, β'2mp and β'2mp bilateral) in a confined area in CRE and SMP. In addition, some MBONs appear to terminate on the dendrites of DANs innervating other compartments, forming feedback loops. Using these mechanisms, local modulation in a specific compartment could broadly impact the ensemble of MBON activity and how it is interpreted.

Testing these and other models for the roles of the MBON network, both within the MB lobes and in the surrounding neuropils, will be facilitated by an EM-level connectome to confirm the synaptic connections we have inferred based on light microscopy. We will also need physiological assays to confirm the sign of synaptic connections and to measure plasticity. For example, we do not know the sign of action of glutamate in the targets of glutamatergic MBONs, as this depends on the receptor expressed by the target cells (*Xia et al., 2005*; *Jan and Jan, 1976*; *Liu and Wilson, 2013*). In this regard, we note that previous studies demonstrated a role for NMDA receptors in olfactory memory (*Wu et al., 2007*; *Miyashita et al., 2012*).

Neurons that are thought to mediate innate response to odors—a subset of projection neurons from the antennal lobes and output neurons from the lateral horn—also project to these same convergence zones (*Figure 15*; *Figure 15—figure supplement 1*). We propose that these convergence zones serve as network nodes where behavioral output is selected in the light of both the innate and learned valences of stimuli. What are the neurons downstream to these convergence zones? One obvious possibility is neurons that project to the fan-shaped body of the central complex whose dendrites are known to widely arborize in these same areas (*Hanesch et al., 1989*; *Young and Armstrong, 2010a*; *Ito et al., 2013*; *Yu et al., 2013*). It would make sense for the MB to provide input to the central complex, a brain region involved in coordinating motor patterns (*Strauss, 2002*). *Figure 15* provides a diagrammatic summary of these proposed circuits.

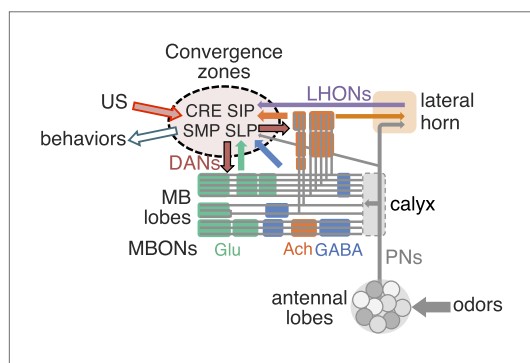

**Figure 15**. Convergence zone of MBON terminals as network nodes to integrate innate and learned valences. MB lobes are consisted of three groups of compartments based on the putative transmitter of MBONs (glutamate, GABA and acetylcholine; color-coded as indicated), which are interconnected inside the lobes and send converging outputs to the lateral horn, CRE, SMP, SIP and SLP. These regions also receive input from the antennal lobes and the lateral horn, some of which appear to be in close apposition to the terminals of MBONs and likely target common downstream neurons (see *Figure 15— figure supplement 1*). Therefore, these convergence zones are well positioned to function as integration sites for selecting adaptive behaviors based on both innate and learned valences.

The following figure supplement is available for figure 15:

**Figure supplement 1**. Convergence of olfactory pathways.

## Dealing with redundancy and resiliency of networks

Using inactivation to uncover the roles of specific cell types is inherently limited by redundancy and resiliency within the underlying neural circuits. For example, consider the MBONs from the α/β lobes. The output of the α/β Kenyon cells is known to be required for retrieval of aversive memory (*Isabel et al., 2004*; *Krashes et al., 2007*; *McGuire et al., 2001*; *Cervantes-Sandoval et al., 2013*). Our anatomical and behavioral results show that MBON-γ1pedc>α/β, a cell type we found to be critical for aversive memory, has terminals largely confined inside the α/β lobes, well-positioned to regulate a total 6 types of MBONs from the α/β lobes (*Figure 1B*). Yet we did not detect a requirement for any of these MBONs in short term aversive memory when tested individually. The ability to detect phenotypes also depends on the strength of the effector; for example, four glutamatergic MBON drivers showed aversive memory impairment in initial screening assays with a strong inhibitor of synaptic function, but we were not able to confirm these effects using a weaker effector (*Figure 6B*).

The failure to see effects when inactivating individual cell types is most easily explained by combinatorial roles and redundancy between MBONs. We note that this high level of resiliency is very reminiscent of observations made with

genetic networks, where less than half of gene knockouts of evolutionarily conserved *Drosophila* genes result in a detectable phenotype (*Ashburner et al., 1999*). Whether or not we detect a requirement for a particular MBON in a particular learning paradigm is likely to depend on which DANs are recruited by the US used in that paradigm as well as the degree of redundancy in the MBON representation of valence. It will be informative to test systematically whether blocking combinations of MBONs, which did not show significant behavioral effect when blocked separately, results in significant memory impairment. It will also be important in future experiments to employ imaging and electrophysiological methods, in which the activities of individual neurons, and the consequences of plasticity, can be observed without being obscured by redundancy.

## Origin of diverse functions of MBONs

The MBs are implicated in functions beyond processing of associative memory (*Martin et al., 1998*; *Liu et al., 1999*; *Joiner et al., 2006*; *Pitman et al., 2006*; *Zhang et al., 2007*; *Hong et al., 2008*). MBONs that influence approach to, or avoidance of, a learned stimulus may also have roles in innate preference behaviors for temperature and hunger-dependent CO2 avoidance (*Hong et al., 2008*; *Bang et al., 2011*; *Bracker et al., 2013*). Moreover, we expect the behavioral repertoire that MBONs govern to go beyond simple approach and avoidance; the MB is known to play a role in experience-dependent regulation of proboscis extension (*Masek and Scott, 2010*) and courtship (*McBride et al., 1999*) as well as regulation of sleep (*Joiner et al., 2006*; *Pitman et al., 2006*) and post-mating behaviors such as oviposition (*Fleischmann et al., 2001*; *Azanchi et al., 2013*). Intriguingly, we found that MBONs whose activation was repulsive promoted wakefulness, whereas MBONs whose activation was attractive promoted sleep; it would make sense for flies to be awake and attentive in an adverse environment. Other internal states, in addition to sleep, are likely to affect the decision to carry out a particular memory-guided behavior; for example, the state of satiety has been shown to regulate memory expression (*Krashes et al., 2009*). The diverse influences of MBONs on behavior can be most easily explained if we assume that the activity of the ensemble of MBON conveys an abstract representation of both valence and internal state. In this view, the ensemble of MBONs may represent internal states along axes such as pleasant-unpleasant or aroused-not aroused. It is upon these axes that primitive forms of emotion are thought to have evolved (*Anderson and Adolphs, 2014*).

# Materials and methods

## Genetic and anatomical methods

The construction and characterization of the split-GAL4 lines are described in detail in (*Aso et al., 2014*). The MBON cell types are listed in *Table 1* and diagrammed in *Figure 1B*. The split-GAL4 lines used in this study are described in *Table 2*; expression patterns, using the most relevant UAS-reporter, are shown in the Figures. *pBDPGAL4U* (*attP2*), an enhancerless GAL4 construct (*Pfeiffer et al., 2010*), was used as a control driver line in behavioral assays.

To combine the expression patterns observed in two split-GAL4 lines, flies were generated by standard genetic crosses that contained the two DNA-binding (DBD) halves and the two activation-domain (AD) halves found in the parent split-GAL4 lines; the AD and DBD components of all split-GAL4 lines are given in *Table 1* of the accompanying manuscript (*Aso et al., 2014*). In general, these lines contained more off-target cells than the parent split-GAL4 lines, due to the interactions of the AD and DBD combinations not present in the parent lines. To identify lines for behavioral experiments, we directly assessed the expression patterns; the majority of combinations produced useful reagents (*Figure 4—figure supplement 1* shows the expression patterns of those combinations used in this work).

The following constructs were used for activating or silencing neuronal function: 5*XUAS-CsChrimson-mVenus* (*attP18*), *10XUAS-CsChrimson-mVenus* (*attP18*), *20XUAS-CsChrimson-mVenus* (*attP18*) (*Klapoetke et al., 2014*); pJFRC124-20XUAS-IVS-dTrpA1 (*attP18*); *10XUAS-dTrpA1* (*attP16*) (*Hamada et al., 2008*); pJFRC100-20XUAS-TTS-Shibire$^{ts1}$-p10 (*VK00005*) (*Pfeiffer et al., 2012*); *UAS-Shi x1* was generated in Thomas Preat's lab by segregating one of the multiple insertions found in the lines described by (*Kitamoto, 2001*).

To compare the expression levels driven by split-GAL4 drivers in specific cell types, 3–7 days post-eclosion female brains were dissected, antibody-stained, mounted and imaged at 20× under identical conditions (see the accompanying manuscript for details) (*Aso et al., 2014*). The relative expression levels in individual cell types are presented as a 0–5 unit gray scale based on the intensity of the signals

**Table 2.** Split GAL4 drivers for MBONs

| Driver | Cell types | p65ADZp DNA | ZpGdbd DNA |
|---|---|---|---|
| MB002B | (MBON-γ5β'2a), MBON-β'2mp | R12C11 | R14C08 |
| MB011B | MBON-γ5β'2a, MBON-β'2mp, MBON-β'2mp_bilateral | R14C08 | R15B01 |
| MB018B | MBON-α'2 | R20G03 | R19F09 |
| MB027B | MBON-α'3m, MBON-α'3ap | R24H08 | R53F03 |
| MB050B | MBON-α2sc, MBON-α'1 | R65B09 | R11F03 |
| MB051B | MBON-γ2α'1, (MBON-α'2) | R70B10 | R19F09 |
| MB052B | MBON-α2sc, (MBON-α2p3p), (MBON-α'3m), MBON-α'3ap, MBON-α'1 | R71D08 | R11F03 |
| MB057B | MBON-β'1 | R80G12 | R53H03 |
| MB077B | MBON-γ2α'1 | R25D01 | R19F09 |
| MB077C | MBON-γ2α'1 | R25D01 | R19F09 |
| MB082C | MBON-α3, MBON-α'2 | R40B08 | R23C06 |
| MB083C | MBON-γ3, MBON-γ3β'1 | R52G04 | R94B10 |
| MB085C | MBON-γ1pedc>α/β | R52H01 | R52B07 |
| MB090C | MBON-γ2α'1, (MBON-α'2) | R73H08 | R19F09 |
| MB091C | MBON-α'2 | R73H08 | R20G04 |
| MB093C | MBON-α3, (MBON-α'2) | R73H08 | R40B08 |
| MB110C | MBON-γ3, MBON-γ3β'1 | R20A02 | R94B10 |
| MB112C | MBON-γ1pedc>α/β | R93D10 | R13F04 |
| MB210B | MBON-γ5β'2a, MBON-β'2mp, (MBON-β'2mp_bilateral) | R15B01 | R27G01 |
| MB242A | MBON-calyx | R64F07 | R57C10 |
| MB262B | MBON-γ1pedc>α/β | R52B07 | R52H01 |
| MB298B | MBON-γ4>γ1γ2 | R53C03 | R24E12 |
| MB310C | MBON-α1 | R52G04 | R17C11 |
| MB399B | MBON-β2β'2a | R21D02 | R22C12 |
| MB433B | (MBON-γ4>γ1γ2), MBON-β1>α | R30E08 | R11C07 |
| MB434B | MBON-γ4>γ1γ2, MBON-β1>α | R30E08 | R53C10 |
| MB438B | PPL1-γ1pedc, (PPL1-α2α'2) | R30E11 | R22B12 |
| MB542B | MBON-α2p3p, (MBON-α'3m), MBON-α'1 | R65B09 | R51D04 |
| MB543B | MBON-α'3m, (MBON-α'3ap), MBON-α'1 | R65B09 | R81E11 |
| MB549C | MBON-α2sc, MBON-α'3ap | R71D08 | R49C12 |
| MB622B | MBON-calyx | R64F07 | R64F07 |

For each of 31 driver lines used in this study, the MBON cell types in which expression is seen as well as the enhancer fragments used for the activation domain (p65ADZp) and DNA binding domain (ZpGAL4DBD) hemi-driver constructs are given. All ZpGAL4DBD constructs were inserted in attP2. The insertion sites of the p65ADZp constructs are indicated by the letter at the end of the driver name as follows: A, *su(Hw)attP8*; B, *attP40*; C, *VK00027*. The cell types shown in brackets indicate expression in those cells was weak, stochastic or was only observed with a subset of UAS-reporters.

in the dendrites obtained for each cell type in each split-GAL4 line. The signal intensity depends on the morphology of individual cell types as well as how many cells of the same cell type innervate the same compartment; thus comparing intensities across cell types is less accurate than the comparisons between lines for the same cell type.

Since the purpose of obtaining these data was to estimate the expression levels of the UAS-effectors used to manipulate cell function, and expression levels are known to vary with genomic insertion site (*Pfeiffer et al., 2010*), we sought to collect expression data using UAS-indicator lines inserted into

the same genomic location as the effectors. We believe that this practice addresses a potential weakness in many prior studies where expression patterns have been determined with an indicator construct inserted in one chromosomal site, while perturbing function with an effector residing at another site. The lack of precise correlation between the expression pattern of the indicator and effector introduces significant uncertainty. The best practice would be to directly measure the expression of the effector protein itself, something we were able to do for the red-shifted channel rhodopsin CsChrimson-mVenus (*Klapoetke et al., 2014*) by staining for mVenus. The next best approach is to have the indicator of expression and the effector inserted at the same chromosomal location, which we were able to achieve for all cases except the assays of the 2 hr aversive and appetitive odor memory which used a weaker UAS-Shibire effector based on a P-element insertion. We used the following UAS-indicators for the matrices shown in Figures: *Figure 2B*, *20XUAS-CsChrimson-mVenus* (*attP18*) reared at 22°C; *Figures 7B, 9B, 10B, 11B*, *pJFRC2-10XUAS-IVS-mCD8::GFP* (*VK00005*) reared at 18°C [similar expression was observed with *pJFRC225-5xUAS-IVS-myr::smGFP-FLAG* (*VK00005*) reared at 25°C]; *Figure 12B*, *pJFRC200-10xUAS-IVS-myr::smGFP-HA* (*attP18*) reared at 22°C. Full confocal stacks of these images are available at www.janelia.org/split-gal4.

We used a set of highly specific GAL4 drivers, made using the split-GAL4 intersectional approach (*Aso et al., 2014*). These drivers have much more restricted expression patterns than those previously used, allowing greater certainty in assigning the effects of perturbations to specific cells. Even with these improved GAL4 drivers, there can be significant variation in expression levels between drivers or in different cell types within the expression pattern of a single driver, as well as off-target expression and variations in genetic background. To assign a function to a cell population, we therefore required that the effects of a manipulation be observed using two different GAL4 drivers for that cell population. In cases where we only had one split-GAL4 driver for a cell type, we believe it is only appropriate to interpret an observed phenotype as suggestive, except in cases where we were confirming a previously published result. Finally, we interpret some results as simply raising the possibility of a role for a cell type. For example, where one GAL4 line resulted in a significant effect, but a second line with a very similar expression pattern did not. There were also cases where we saw a consistent tendency in multiple lines, but where none of the individual lines themselves reached statistical significance.

Detailed methods for immunohistochemistry and image analysis are described in the accompanying manuscript (*Aso et al., 2014*). For the data in *Figure 6—figure supplement 1B*, rabbit anti-GABA (1:500; A2052, Sigma-Aldrich, St Louis, MO 63103) was used.

## Assessing innate behaviors

We asked if using the split-GAL4 lines to activate or inactivate neurons would perturb general innate behaviors, such as locomotion and visual perception that might interfere with our assays of memory, locomotion and sleep. Given the results of these tests, we selected 23 split-GAL4 lines for use in the primary behavioral screening of the MBONs and additional lines to confirm the results of primary screening (see *Table 1*). We also performed behavioral assays to verify that the animals carrying the driver line and effector were able to perceive odors, electric shocks and sugar rewards (see Results). Thus, the behavioral phenotypes we observed in these lines are unlikely to result from general defects in innate behaviors.

We first screened 27 MBON split-GAL4 driver lines, crossed to a multi-insert UAS-Shibire[ts1] effector line (*UAS-Shi[ts1]* on the third chromosome) (*Kitamoto, 2001*) at 34°C. 3–7 days old adult males of each genotype were wet starved for 1–4 hr and then were assayed for 22 parameters of basic locomotion in response to startle, optomotor and phototaxis stimuli using an apparatus (Fly Behavioral Olympiad, unpublished) inspired by published assays (*Benzer, 1967*; *Zhu et al., 2009*). Fourteen split-GAL4 lines showed a difference in one or more parameters from the pBDPGAL4U and these lines were re-screened with *pJFRC100-20XUAS-TTS-Shibire-ts1-p10* (*VK00005*) DL (*Pfeiffer et al., 2012*).

Although some lines had phenotypes in some behavioral categories, none of the output lines showed consistent phenotypes across the two Shibire effectors. All of the lines screened were able to appropriately respond to visual stimuli, showed positive phototaxis towards green and UV light, and were able to walk when MBONs were inactivated with Shibire. Only one line, MB549C, showed a significant reduction in walking speed, though subsequent analysis suggests this reduction in speed was due to the genetic background of that line rather than silencing of MBONs. Thus flies can move and orient themselves when MBONs, and any neurons with off-target expression in these split-GAL4

drivers, are inactivated and the small differences from wild-type would not be expected to significantly affect behavior in the assays we performed that used Shibire as an effector.

We also assayed the behavior of flies from each split-GAL4 driver line in a high-throughput open-field arena described in (*Kabra et al., 2013*) for 15 min during dTrpA1 activation using *10X UAS-dTRPA1* (*attP16*) (*Hamada et al., 2008*) at 30°C. We first tracked the body and wing position of the flies (*Branson et al., 2009*; *Kabra et al., 2013*), and then automatically annotated 14 social and loco-motor behaviors of flies such as walking, chasing, grooming, etc (*Supplementary file 1*) (*Kabra et al., 2013*). Although we observed variation of locomotion levels between drivers and their GAL4/+ controls, only one driver, MB052B/dTrpA1, showed an obvious phenotype. This phenotype of MB052B was limited to male flies and presumably attributed to male specific expression in this line; we therefore did not use males of this driver in experiments involving activation.

## Optogenetics

The choice assay was performed in a 10 cm diameter and 3 mm high circular arena as previously described (*Klapoetke et al., 2014*). Flies expressing CsChrimson were allowed to distribute between two dark quadrants and two quadrants illuminated with 617 nm LEDs (Red-Orange LUXEON Rebel LED—122 lm; Luxeon Star LEDs, Brantford, Ontario, Canada). This wavelength efficiently activates neurons expressing CsChrimson (*Klapoetke et al., 2014*), but was distant enough from the peak absorption spectrum of endogenous rhodopsins that at the light intensity used (34 µW/mm$^2$) negligible phototaxis of the control genotype was observed. To maintain a constant temperature, the LED board was placed on a heat sink and air (150 ml/min) was exchanged through holes at the center and four corners of the arena in a similar way as in the previously described olfactometer (*Vet et al., 1983*). The four quadrants were separated by 1 mm dividers. The bottom of arena consisted of a 3 mm thick diffuser with an IR absorption film (YAG, Laser PVC Film; Edmund optics, Barrington, NJ 08007-1380). The intensity of red light decreased from 34 to 3 µW/mm$^2$ over a 10 mm gradient extending from the border between light on and off quadrants, as shown in *Figure 5A*.

Crosses were kept on standard cornmeal food supplemented with retinal (0.2 mM all-*trans*-retinal prior to eclosion and then 0.4 mM) at 22°C at 60% relative humidity in the dark. Groups of approximately twenty 4–10 days post-eclosion females were tested at 25°C at 50% relative humidity in a dark chamber. Videography was performed under reflected IR light using a camera (ROHS 1.3 MP B&W Flea3 USB 3.0 Camera; POINT GREY, Richmond, BC, Canada) with an 800-nm long pass filter (B&W filter; Schneider Optics) at 30 frames per sec,1024 × 1024 pixel resolution and analyzed using Fiji (*Schindelin et al., 2012*) or Ctrax (*Branson et al., 2009*). Statistical comparisons were performed using Prism (Graphpad Inc, La Jolla, CA 92037); Kruskal Wallis One way ANOVA followed by Dunn's post-test for comparison between control and experimental genotype in *Figure 2C* and *Figure 5—figure supplement 1*; One way ANOVA followed by Bonferroni's multiple comparison test for *Figures 3G and 4*; In *Figure 5D*, p-values for the exit direction were computed using the test of equal proportions from R (http://stat.ethz.ch/R-manual/R-patched/library/stats/html/prop.test.html) followed by multiple comparisons with the Dunn-Sidak correction. Only data obtained with *20XUAS-CsChrimson-mVenus* (*attP18*) are shown in this study. Our preliminary results with 5*XUAS-CsChrimson-mVenus* (*attP18*) or *10XUAS-CsChrimson-mVenus* (*attP18*) indicate that either too weak or too strong expression may result in a failure to observe a phenotype.

## Aversive (shock) and appetitive (sucrose) 2-hour odor memory

Behavioral experiments were performed at 60% humidity in dim red light for training and in complete darkness for test. The odors, 3-octanol (OCT; Merck) and 4-methylcyclohexanol (MCH; Sigma–Aldrich) were diluted to 1% and 2%, respectively, in paraffin oil (Sigma–Aldrich). Flies were placed in the apparatus and shifted to the restrictive temperature (32°C) from 30 min prior to the commencement of training until the end the experiment. During the 2-hr period between training and testing, trained flies were kept in a vial with moistened filter paper. The trained flies were then allowed to choose between MCH and OCT for 2 min in a modified transparent T-maze. Odors were placed in cups with 30 mm diameter and delivered at a flow rate of 0.6 l/min per tube. The distribution of the flies was monitored by videography and the preference index was calculated by taking the mean indices of the last 10 s in the 2-min choice period. Half of the trained groups received reinforcement together with the first presented odor and the other half with the second odor to cancel the effect of the order of reinforcement.

For aversive memory, a group of ~50 flies in a training tube alternately received OCT and MCH for 1 min in a constant air stream; twelve 1.5 s 90 V electric shocks spaced over 60 s were paired with one of the odors. In the primary screening using *pJFRC100-20XUAS-TTS-Shibire-ts1-p10* (*VK00005*) as the effector, flies were raised at 25°C. In the secondary screening using the *UAS-Shi x1* effector, flies were raised at 18°C. Odor avoidance was measured by asking flies to choose between air and either MCH or OCT at the same concentrations used in the memory assay; these odors are aversive to naïve flies. For shock avoidance, flies were asked to choose between two tubes, both with copper grids, but only one electrically active. For appetitive memory, the conditioning protocol was as described previously (*Liu et al., 2012*). Flies were starved prior to the experiments until ~10% mortality was reached. For sugar attraction, flies are asked to choose between two tubes, one with plain filter paper and one with sugar-embedded paper.

Statistical analyses were performed with Prism5 software (GraphPad). The tested groups that did not violate the assumption of normal distribution (D'Agostino-Pearson test) or homogeneity of variance (Bartlett's test) were analyzed with parametric statistics: one-sample t-test or one-way analysis of variance followed by planned pairwise multiple comparisons (Bonferroni). The significance level for statistical tests was set to 0.05. As some of the data points in *Figure 7B* violated the assumption, non-parametric statistics were applied to the dataset (Kruskal–Wallis test followed by Dunn's multiple test).

## Visual aversive (shock) and appetitive (sucrose) conditioning

The effector-line *pJFRC100-20XUAS-TTS-Shibire-ts1-p10* (*VK00005*) was used, and all experimental flies were heterozygous ($w^+/w$) or wild-type ($w^+/Y$) for *white*. Flies were sorted by genotype under $CO_2$ anesthesia at least 2 days prior to experiments; each measurement used 30–40 mixed males and females. For appetitive conditioning experiments, 2–4 days post-eclosion flies were starved on moistened filter paper to approximately 20% mortality (*Schnaitmann et al., 2010*); for aversive conditioning experiments, flies were not starved. Control responses to sugar and shock were measured as described previously (*Schnaitmann et al., 2010*; *Vogt et al., 2014*).

Appetitive and aversive conditioning paradigms and behavioral tests were as previously described (*Schnaitmann et al., 2013*; *Vogt et al., 2014*). Briefly, conditioned stimuli were presented from below using LEDs with peak wavelengths of 452 nm and 520 nm (Seoul Z-Power RGB LED) or 456 nm and 520 nm (H-HP803NB, and H-HP803PG, 3 W Hexagon Power LEDs, Roithner Lasertechnik, Vienna, Austria), adjusted to 14.1 Cd m$^{-2}$ s$^{-1}$ (blue) and 70.7 Cd m$^{-2}$ s$^{-1}$ (green). Each quadrant of the arena was also equipped with an IR-LED (850 nm) to provide background illumination for videography. For appetitive conditioning, filter paper soaked in 2 M sucrose and subsequently dried was presented as reward (*Schnaitmann et al., 2010*). For aversive conditioning, a 1 s electric shock (AC 60 V) was applied 12 times in 60 s during CS + presentation using a transparent shock grid made of laser-structured ITO on a glass plate. In both appetitive and aversive conditioning assays, differential training was followed by a binary choice without reinforcement. During the 90 s test period, blue and green light were presented in two diagonally opposite quadrants of the arena and the color choice of flies was recorded from above at 1 frame per second with a CMOS camera (Firefly MV, Point Grey). A preference index for each frame was calculated by subtracting the number of flies on the green quadrants from the number on the blue quadrants, divided by the total number of flies. The difference in average visual stimulus preference between the two groups was used to calculate a performance index. Sugar preference and shock avoidance tests were performed as described previously (*Schnaitmann et al., 2010*; *Vogt et al., 2014*).

Statistical analyses were performed with Prism5 software (GraphPad). The groups that did not violate the assumption of normal distribution (Shapiro–Wilk test) or homogeneity of variance (Bartlett's test) were analyzed with parametric statistics: one-way analysis of variance followed by the planned pairwise multiple comparisons (Bonferroni). For data that significantly differed from the normal distribution or did not show homogeneity of variance (Bartlett's test), non-parametric statistics were applied (Kruskal–Wallis test followed by Dunn's multiple test). The significance level of statistical tests was set to 0.05. Only the most conservative statistical result of multiple pairwise comparisons is indicated.

## Aversive (shock) long-term odor memory

For behavioral experiments concerning long-term aversive olfactory memory (LTM), wild type (Canton S) or Split-GAL4 female flies were crossed to either *pJFRC100-20XUAS-TTS-Shibire-ts1-p10* (*VK00005*) (outcrossed to a Canton S genetic background) or wild type males. Flies were raised on standard

medium containing yeast, cornmeal and agar at 18°C and 60% relative humidity under a 12 hr:12 hr light–dark cycle.

The day before the experiment, 0–2 days post-eclosion flies were transferred to fresh food vials. Flies were trained with five cycles of aversive conditioning spaced by 15 min inter-trial intervals (spaced conditioning) at 25°C. The time course of one cycle of aversive conditioning was as follows: flies were exposed to the first odorant for 1 min while twelve 1.5 s, 60-V electric shocks, separated by 3.5 s, were delivered; after a 45 s rest, flies were exposed to the second odorant for 1 min. Two odorants, 3-octanol and 4-methyl-cyclohexanol, were used alternatively as the conditioned stimulus. For all assays (training, memory test and olfactory acuity), odorants were diluted in mineral oil at a final concentration of 0.36 mM for octanol and 0.325 mM for methyl-cyclohexanol, and were delivered by 0.4 l/min airflow bubbled through odor-containing bottles. Except during conditioning, flies were kept on food and were maintained at 18°C between training and test. The memory test was performed as described in (*Trannoy et al., 2011*). Flies were allowed to acclimatize to the restrictive (32°C) or permissive (25°C) temperature for 30 min prior to the test. Memory scores are displayed as means ± SEM.

In a primary screen, each Split-Gal4 line was tested for aversive LTM (n = 7-10, except MB093C, n = 20). The scores obtained for each line were compared by a two-tailed unpaired t-test to the pool of +/UAS-Shi$^{ts}$ scores (n > 150). Due to multiple comparisons, a Benjamini-Hochberg procedure (*Benjamini and Hochberg, 1995*) was applied to control the false discovery rate with a significance level of 0.05. Putative hits after the primary screen were then re-assayed in comparison with +/UAS-Shi$^{ts}$ and Split-GAL4/+ at both the restrictive and permissive temperatures. In this second set of experiments, the scores from the three genotypes were compared using one-way ANOVA followed by pairwise comparisons by Newman–Keuls posthoc tests. MB052B/Shi showed memory impairment compared to MB052B/+ and +/Shi at the restrictive temperature (ANOVA, $F_{2,33}$ = 14.84, p < 0.0001; ***: p < 0.001 by Newman–Keuls pairwise comparison; $n \geq 9$ for all genotypes) but not at the permissive temperature (ANOVA, $F_{2,24}$ = 0.95, p = 0.40, $n \geq 8$ for all genotypes).

## Memory for ethanol intoxication

Flies were reared at room temperature (22°C–23°C) under ambient light with no constrained light/dark cycle. Split-GAL4 males were crossed to *pJFRC100-20XUAS-TTS-Shibire-ts1-p10* (*VK00005*) females. A daily control (*pJFRC100* x *pBDPGAL4U*) was run alongside all experimental crosses. Split-GAL4/+ crosses were performed at a different time than the original split-GAL4/Shi$^{ts}$ screen and run alongside UAS-Shi$^{ts}$/+ flies. Thermoactivation of Shi$^{ts1}$ was carried out at 31°C during both the training and test period. For 24-hr memory, flies were kept at 22°C–23°C under ambient room light between training and test.

Odors used were 1:36 (vol:vol) ethyl acetate in mineral oil and 1:36 (vol:vol) iso-amyl alcohol in mineral oil. Choice tests for groups of 30 flies were performed in a Y-maze (each arm 2.5 cm in length and 1.5 cm in diameter). Odors were actively streamed individually through the top arms of the Y at 0.3 l/min. Vials of flies were placed at the lower Y arm and flies climbed up and chose between opposing arms of the Y into 14 ml culture tubes; one arm contained one of the odors and the other arm contained air streamed through mineral oil. The preference index was calculated by the formula: (number of flies in odor vial–number of flies in air vial) / total number of flies. The conditioned preference index was the average of the preference indices in reciprocal trials.

Ethanol conditioning was performed essentially as described in (*Kaun et al., 2011*). Groups of 30 males were trained in perforated 14 ml culture vials filled with 1 ml of 1% agar and covered with mesh lids. 96 vials of flies were trained simultaneously in two 30 × 15 × 15 cm training boxes. Training consisted of a 10 min habituation to the training chamber with air, a 10 min presentation of odor 1 (1:36 odor:mineral oil actively streamed at 2 l/min), then 10 min of odor 2 (1:36 odor:mineral oil actively streamed at 2 l/min), with 60% ethanol. Air flow was matched for CS$^+$ and CS$^-$ experiments, and ethanol was delivered by mixing pure ethanol vapor (1.5 l/min) with humidified air (1.1 l/min) at a specified ratio (*Wolf et al., 2002*). Reciprocal training was performed to ensure that an inherent preference for either odor did not affect the results. Vials of flies from Group 1 and Group 2 were paired according to placement in the training chamber and tested simultaneously. Flies were tested in the Y-maze described above either 30 min or 24 hr after training. Reciprocal groups were averaged for each n = 1.

Statistical analyses were performed using the statistical software JMP 10.0.0 (SAS Institute, Inc., Cary, NC 27513-2414). Each split-GAL4/UAS-Shi$^{ts1}$ cross was run on two separate days and pooled

for a total with n = 12/group. Statistical significance for any split-GAL4 line was determined by performing a Wilcoxon test for each split-GAL4/UAS-Shi[ts] cross against a pooled control. The pooled control included 12 randomly sampled means from the *pBPDG4U*/UAS-Shi[ts] daily control. A Benjamini-Hochberg False discovery rate (FDR) test (*Benjamini and Hochberg, 1995*) was performed on the p-values for each Wilcoxon comparison. Lines showing $p < 0.05$ following the FDR test were considered statistically significant. GAL4/+ controls were performed for these significant hits, and Kruskal–Wallis comparisons were made comparing each split-GAL4/UAS-shi[ts], split-GAL4/+ and +/UAS-Shi[ts]. Lines were considered significant hits if they passed the FDR correction, splitGAL4/+ Kruskal–Wallis test and showed no significantly decreased sensitivity for either odor used in the assay.

## Sleep

Split-GAL4 flies were crossed to either *10X UAS-dTrpA1* (*attP16*) (*Hamada et al., 2008*) or *pJFRC124-20XUAS-IVS-dTrpA1* (*attP18*) and maintained at 21–22°C. Virgin female progeny, 3–7 days post-eclosion, (n = 24–35) were placed in 65 mm × 5 mm transparent plastic tubes with standard cornmeal dextrose agar media, placed in a *Drosophila* Activity Monitoring system (Trikinetics) and locomotor activity data were collected in 1-min bins for 7 days. Activity monitors were maintained with a 12 hr:12 hr light–dark cycle at 65% relative humidity. Total 24-hr sleep amounts (daytime plus nighttime sleep) were extracted from the locomotor data as described by (*Donelson et al., 2012*); sleep was defined as 5 min or more of inactivity (*Hendricks et al., 2000*; *Shaw et al., 2000*). Sleep profiles were generated representing average (n = 24–32) sleep (min/30 min) for day 3 (baseline), days 4 and 5 (activation), and day 6 (recovery). In addition to permissive temperature controls, *pBDPGAL4U /dTrpA1* and *split-GAL4/+* were used as genotypic controls for hit detection. For all screen hits, waking activity was calculated as the number of beam crossings/min when the fly was awake; consistent with the assays performed in the Flybowl, none of the lines had discernable locomotor defects. Statistical comparisons between experimental and control genotypes were performed using Prism (Graphpad Inc) by Kruskal Wallis One way ANOVA followed by Dunn's post-test.

## Acknowledgements

We thank Sung Soo Kim, TJ Florence, Michael Reiser, Igor Negrashov, Steven Sawtelle, Jinyang Liu for help in establishing the optogenetics apparatus. The Janelia Scientific Computing Software group generated software tools, Brandi Sharp and the Janelia Fly Facility helped in fly husbandry, Susana Tae, Rebecca Vorimo and the FlyLight Project Team performed brain dissections, histological preparations and confocal imaging and Heather Dionne made several molecular constructs. We thank Fabian Stamp and Megan Atkins for assistance in visual learning and optogenetics assays, respectively. We thank Brett Mensh, Joshua T Dudman, Vivek Jayaraman, Larry F Abbott, Daisuke Hattori, Toshide Hige and Glenn Turner for stimulating discussions and for comments on earlier drafts of the manuscript.

## Additional information

### Funding

| Funder | Grant reference number | Author |
|---|---|---|
| Howard Hughes Medical Institute | | Yoshinori Aso, Karla R Kaun, Alice A Robie, William J Rowell, Rebecca M Johnston, Teri-T B Ngo, Nan Chen, Wyatt Korff, Ulrike Heberlein, Kristin M Branson, Gerald M Rubin |
| Howard Hughes Medical Institute | Janelia Visiting Scientist Program | Yoshinori Aso, Divya Sitaraman, Katrin Vogt, Michael N Nitabach, Hiromu Tanimoto |
| Agence Nationale de la Recherche | | Pierre-Yves Plaçais, Thomas Preat |

| Funder | Grant reference number | Author |
|---|---|---|
| Labex MemoLife | | Ghislain Belliart-Guérin |
| Bundesministerium für Bildung und Forschung | Bernstein Focus Neurobiology of Learning 01GQ0932 | Hiromu Tanimoto |
| Max-Planck-Gesellschaft | | Hiromu Tanimoto |
| Deutsche Forschungsgemeinschaft | TA 552/5-1 | Hiromu Tanimoto |
| Japan Society for the Promotion of Science | MEXT/JSPS KAKENHI 25890003, 26120705, 26119503 and 26250001 | Hiromu Tanimoto |
| Naito Foundation | | Hiromu Tanimoto |
| National Institute of Neurological Disorders and Stroke | R01NS055035, R01NS056443, R01NS091070 | Michael N Nitabach |
| National Institute of General Medical Sciences | R01GM098931 | Michael N Nitabach |

The funders had no role in study design, data collection and interpretation, or the decision to submit the work for publication.

### Author contributions

YA, Conceived and designed the overall study, conducted optogenetics assays, analyzed and interpreted data, wrote the article; DS, Conducted, analyzed and interpreted assays of sleep regulation; TI, Conducted, analyzed and interpreted assays of short-term olfactory memory; KRK, Conducted, analyzed and interpreted assays of ethanol-reward memory; KV, CS, Conducted, analyzed and interpreted assays of visual memory; GB-G, P-YP, Conducted, analyzed and interpreted assays of long-term olfactory memory; AAR, Conducted, analyzed and interpreted the Flybowl assay; NY, Conducted, analyzed and interpreted the US-substitution experiment for short-term olfactory memory; WJR, Conducted, analyzed, and interpreted assays of the optomotor behavior; RMJ, Prepared samples and acquired confocal images; T-TBN, Conducted optogenetic assays; NC, Conducted sleep regulation assays; WK, Analyzed and interpreted assays of optomotor behavior; MNN, Analyzed and interpreted assays of sleep regulation; UH, Analyzed and interpreted assays of ethanol-reward memory; TP, Analyzed and interpreted assays of long-term olfactory memory; KMB, Analyzed and interpreted the results of the optogenetic and Flybowl assays; HT, Conceived and designed the overall study and analyzed and interpreted assays of short-term olfactory and visual memory; GMR, Conceived and designed the overall study, interpreted data and wrote the article

### Author ORCIDs

Wyatt Korff, http://orcid.org/0000-0001-8396-1533

## Additional files

### Supplementary file

• Supplementary file 1. Significance of behavioral differences for Fly Bowl assay. For each line, we compared the fraction of time the split-GAL4/dTrpA1 line does each of 14 behaviors to two controls: empty GAL4 crossed to dTrpA1 (pBDPGAL4U/dTrpA1) and split-GAL4/+. Each data point corresponds to the fraction of time that all 20 flies in a given video performed the given behavior. The table contains a dash if the behavioral differences to both controls had different signs. Otherwise, we use the Wilcoxon rank-sum test to compare to each control separately, adjust the p-value using the Benjamini-Hochberg correction for controlling the false discovery rate to 0.05, and report the (conservative) maximum adjusted p-value of the comparison to each control. The average number of videos (n) analyzed for each split-GAL4/TrpA1 line was 5.0, for each split-GAL4/+ line was 4.2, and for the pBDPGAL4U/TrpA1 line was 143.

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
