## [Decision Letter]

Thank you for sending your work entitled ”Mushroom body output neurons encode valence and guide memory-based action selection in *Drosophila*“ for consideration at *eLife.* Your article has been favorably evaluated by K VijayRaghavan (Senior editor) and 3 reviewers, two of whom is a member of our Board of Reviewing Editors.

The following individuals responsible for the peer review of your submission have agreed to reveal their identity: Liqun Luo (Reviewing editor); Leslie Griffith, Yi Zhong, and Josh Dubnau (peer reviewers).

The Reviewing editor and the other reviewers discussed their comments before we reached this decision. The Reviewing editor has included below the comments from three reviewers. The reviewers are impressed by the study, and most comments are meant to help further improve/clarify the presentation and can be address by textual changes. We hope that your revised manuscript will address these points.

Reviewer 1:

This paper provides the first behavioral look at the function of newly identified MBONs. The authors (and they are multitude, reflecting the need for expertise for each behavior examined) suggest the major function of MBONs is to encode valence. In this view MB outputs are not themselves driving motor patterns, but rather they are setting the bias to favor some subset of the possible motor programs. The support for this idea is necessarily at this point somewhat circumstantial, but the upshot of the behavioral experiments is that many different behaviors require the exact same MBON or set of MBONs. The commonality between the behavioral situations does not have to do with the qualities or sensory mode of the stimulus, but rather with whether or not the animal likes or dislikes it. Importantly, the isolated activation of these neurons can be shown, in some cases, to appear to drive the animal to behave as if it has an aversion or attraction. The small number of MBONs coupled with the large number of MB-influenced behaviors demand this type of model, since one-to-one connections with motor systems would simply be anatomically untenable.

I have no substantive negatives. Overall, I think that this is an incredibly important study for several reasons. First, it is a catalog of the functional anatomy of MBONs, laying out the first pass results of what happens in several behavioral contexts when you manipulate their activity. This will be an important resource for many in the community. Second, and critically, this paper goes beyond cataloging to give us a credible model of what this piece of brain is built to do. This is something that will be important for anyone working in any nervous system who is interested in how behavioral decisions are made.

The way the paper was written also deserves comment. It is very dense and very broad ranging due to the sheer number of experiments done. In spite of this, the paper manages to very clearly deliver its bottom lines, keeping the reader's eye on the message during the first read through. The details are all there when you are ready to go back to look at them, but they don't distract. This will allow a wide range of readers to access this paper and actually get something out of it. I also like the ”Rosetta Stone“ in Table 1 that provides orientation necessary to integrate this study with the previous literature.

Reviewer #2:

This is a beautiful work that attacks most fundamental but highly difficult problem in the field of neuroscience: how valence is encoded within a neural network and how memory is associated with such organization. It fully takes advantage of accessibility to powerful genetic tools available to *Drosophila*. The proposed model is insightful and supported by a large body of data.

*Reviewer #3*:

First, I want to say that this is a magnum opus that will become an instant classic in this field. This is the sort of paper that we have all expected to see come out of the Janelia effort in flies. It is the sort of study where the impact is even greater than would be expected by addition of the parts.

The authors took a systematic approach, a systems biology approach, to attack what is really the next big question in the ”mushroom body field“: what do mushroom bodies actually do. A huge body of literature coming from several different insect species (not just *Drosophila*; maybe check to see that some of the other insects are cited early on?) has already established that mushroom bodies are a multi-modal integration center. In *Drosophila*, the bulk of the work on the question “what do MBs do” has come from dissection of olfactory associative memory. In this paper, and accompanying manuscripts, the authors take a big step towards answering this question by systematically cataloging and then manipulating the MB output neurons. After generating reagents to manipulate each of the individual MBONs, the effects of these manipulations were then measured in the context of a series of well-studied MB dependent behaviors. These include innate responses to odor (including attraction vs avoidance), aversive and appetitive odor memory, aversive and appetitive visual learning, aversive and appetitive ethanol learning, and sleep.

There is of course an impressive amount of effort that underlies this study. But what really makes this manuscript compelling for a high impact journal is that the findings are synthetic. Conceptual themes emerge that would not be apparent if one studied a subset of MBONs or only one behavior. What emerges is an over-arching hypothesis in which MBs integrate information about the environment (including internal state and past experience) and where activity in individual groups of MBONs bias responses towards outputs that are adaptive given the valance of the stimuli. And these findings hang together for the most part in terms of how MBONs group together in terms of valence in the innate responses, impact on memory, and neurotransmitter types.

On one level, this model is basically how we might have all imagined this could work. But to see the model emerge elegantly from an exhaustive data set is startling nevertheless. Moreover, this study provides starting points for countless follow up studies. Specific hypotheses need to be tested, but this study provides a roadmap and a toolbox to do that. So this will have immense impact on the field.

Overall, I am really enthusiastic about this manuscript. All of my critiques are subordinate to the above positive comments.

Specific critiques (combining all reviewers):

1) Blocking MB output has an all or none impact on memory retrieval. But blocking individual MBONs has smaller impact. This is expected, and is discussed. But there are cases where the added impact of blocking individual MBONs doesn’t add up to the whole behavior. The most striking example is aversive olfactory memory, where only one MBON (gamma-ped>alpha/beta) has an effect, and the effect is partial. This can be explained of course by the potential that combinations of other MBONs can impact (i.e. redundancy), or conceptually at least by there being unknown MBONs. This should be discussed.

2) There is a significant literature on the roles of each of the major MB KC subsets. This consists of two types of experiment: genetic rescue experiments in which individual mutations are rescued with spatially restricted transgene expression, and experiments in which individual KC classes are silenced/activated. The findings with MBONs here are not really integrated into in the Discussion section. The most clear example of this is the MBON-gamma-ped>alpha/beta neuron. In aversive olfactory conditioning, all relevant DA input to DopR comes in through gamma lobes during learning, and retrieval and consolidation involve alpha/beta. So this MBON is ideally positioned to play an important role, but this wasn’t discussed in the text. On the other hand, in appetitive learning, there is evidence (actually from one of the authors) that early and later memories are parallel in gamma and alpha/beta. Again, this is not really discussed. I realize that the manuscript is long, has many findings, and it is not possible to place every finding in context. But this MBON is one of the few that is front and center in this study, so I thought these findings should be integrated better conceptually into the literature.

3) Related to above, in general, the role of individual lobes, the sequential use of MB lobes, the role for ongoing neural activity (prime lobes, APL, DPM) are not really incorporated into the framework provided in the discussion. Since some of the MBONs provide feedback (which is mentioned), i thought this should be better/more explicitly framed.

4) Although the MBON anatomy is documented in a different study, there are many cases in this manuscript where MBON labeling in individual panels (mostly in supplement) are almost impossible to see. I think the authors should provide insets with higher magnification b/c many of the neurons are too small to see when the entire CNS is shown.

5) In the description of Figure 9 findings, it is stated that naive responses to sugar are normal at restrictive temperature for all the lines. When I look at the figure, it appears that it is true that none are different from control, but some lines are higher and some lower than control, and my guess is that the higher lines are significantly different from the lower lines. This is a blemish in the data. Of course all papers have some blemishes, but I think the authors should state the statistical difference if there is one and just make the case that it is not a major concern.

6) In the sleep section, there is no rebound effect after 2 nights of sleep loss (in the lines that reduce sleep). Why?

7) This study crosses many sub-fields, so citation is really not trivial to organize. But I see a few omissions where previous work was not emphasized enough (IMHO). Including these could improve the Discussion section, and will avoid inadvertently annoying a few colleagues:

Citation omissions/mis-citations:

In my opinion, Marin et al. Cell 2002 and Lin et al., Cell 2007 should be cited alongside Vosshall and stocker, 2007; [146]; [136]; [61]. Another place where I see missing citations is where the authors first introduce the aversive and appetitive olfactory assays, but they only cite aversive (Tully Quinn, and some reviews). The appropriate appetitive citations should be included (Tempel original paper, and the modern versions from Waddell and Preat labs).

Previous work not emphasized enough:

I see two corners of the literature that IMHO deserve more attention: First, the findings on glutamatergic MBONs could be discussed in the context of the published role of NMDA receptors in MB dependent learning. There are a series of studies (Saito lab, Chiang/Tully collaboration) on NMDAr that I always found hard to incorporate into a model. But now one might view these in a new light.

The other literature that I think is given short shrift is a bulk of data on the effects of internal state (satiety) on reward learning via monoaminergic modulation of MB. These papers, mostly from Waddell lab, have a major impact on how we frame and think about the current datasets and I imagine they impacted experimental design quite a bit. It might be nice to shed a bit more textual light on this.

8) Concentrations of odors for learning and for task relevant sensorimotor responses not always mentioned. Please add this info and state whether the concentrations for olfactory acuity and learning are the same.

---

## [Author Response]

*1) Blocking MB output has an all or none impact on memory retrieval. But blocking individual MBONs has smaller impact. This is expected, and is discussed. But there are cases where the added impact of blocking individual MBONs doesn’t add up to the whole behavior. The most striking example is aversive olfactory memory, where only one MBON (gamma-ped>alpha/beta) has an effect, and the effect is partial. This can be explained of course by the potential that combinations of other MBONs can impact (i.e. redundancy), or conceptually at least by there being unknown MBONs. This should be discussed*.

As we discussed in the first section of Results and the section of Discussion “Dealing with redundancy and resiliency of networks”, our experimental protocols are likely to result in false negatives. There are several reasons for this. Most importantly, the effects of blocking the small number of cells in individual lines might not rise above the inherent noise of behavioral assays to become statistically significant. Consider the case of olfactory aversive learning pointed out by the reviewer. One neuron, a GABAergic MBON, had a strong effect, but still only resulted in 40-50% reduction compared to the controls. However, in the initial screening with a strong effector (Figure 6), we observed that four glutamatergic MBONs lines had an effect in our initial screening assays, but these same lines did not give a significant effect when subsequently retested with a weaker effector. The simplest explanation of these observations is that these individual MBONs had effects that were only obvious when using a strong effector. Further experiments will be required, but we suspect that these lines do play a role in aversive memory and in combination would produce significant effects with either effector.

We added the following two sentences to highlight the possible role of glutamatergic MBONs in aversive memory:

“The ability to detect phenotypes also depends on the strength of the effector; for example, four glutamatergic MBON drivers showed aversive memory impairment in initial screening assays with a strong inhibitor of synaptic function, but we were not able to confirm these effects using a weaker effector (Figure 6).”

Similarly in the Chrimson activation assays and in long term aversive odor memory, lines expressing in a combination of several cell types gave significant effects but lines for the individual cell types did not. The negative data could be due to too weak or too strong expression of the effector as well as off-targeted expression. To explore such possibility, we have repeated the choice assay of 7 driver lines using 5x, 10x, 20xUAS-CsChrimson in attP18. Presumably depending on the intensity of driver, we observed much weaker or no significant phenotype with 5xUAS-CsChrimson for majority of lines. Conversely, in some driver lines, we observed significant effect with 10xUAS-CsChrimson but not with 20xUAS-CsChrimson. This is possibly due to toxicity of 20xUAS-CsChrimson when expressed at too high level; CsChrimson driven with pan-neuronal drivers such as elav-GAL4 and R57C10-GAL4 are lethal. Because these experiments are preliminary we did not add them to the paper, but did include the following sentence in Materials and Methods: “Only data obtained with 20x*20XUAS-CsChrimson-mVenus* (*attP18*) are shown in this study. Our preliminary results with 5*XUAS-CsChrimson-mVenus* (*attP18*) or *10XUAS-CsChrimson-mVenus* (*attP18*) indicate that either too weak or too strong expression may result in the failure to observe a phenotype.”

It is also formal possibility that we are missing the additional MBONs, but we think this is unlikely (see anatomy paper).

*2) There is a significant literature on the roles of each of the major MB KC subsets. This consists of two types of experiment: genetic rescue experiments in which individual mutations are rescued with spatially restricted transgene expression, and experiments in which individual KC classes are silenced/activated. The findings with MBONs here are not really integrated into in the Discussion section. The most clear example of this is the MBON-gamma-ped>alpha/beta neuron. In aversive olfactory conditioning, all relevant DA input to DopR comes in through gamma lobes during learning, and retrieval and consolidation involve alpha/beta. So this MBON is ideally positioned to play an important role, but this wasn’t discussed in the text. On the other hand, in appetitive learning, there is evidence (actually from one of the authors) that early and later memories are parallel in gamma and alpha/beta. Again, this is not really discussed. I realize that the manuscript is long, has many findings, and it is not possible to place every finding in context. But this MBON is one of the few that is front and center in this study, so I thought these findings should be integrated better conceptually into the literature*.

We added the following sentence to highlight previous studies of dopamine receptor rescue:

“Consistent with these observations, restoring expression of the D1-like dopamine receptor specifically in the γ Kenyon cells has been shown to rescue the aversive odor memory defect of a receptor mutant (106).”

We believe the following existing sentences address the reviewer’s request for discussion of the fact that “In aversive olfactory conditioning, all relevant DA input to DopR comes in through gamma lobes during learning, and retrieval and consolidation involve alpha/beta.”

“One attractive model is that requirement of cholinergic MBONs originates from the transfer of information between disparate regions of the MB lobes through the inter-compartmental MBONs connections within the lobes or by way of connections outside the MB, like those described in the next two sections.”

“The multilayered arrangement of MBONs (see Figure 17 of the accompanying manuscript) (Aso et al., submitted) provides a circuit mechanism that enables local modulation in one compartment to affect the response of MBONs in other compartments.”

“The GABAergic MBON-γ1pedc>α/β and the glutamatergic MBON-β1>α both project to α2 and α3, where their axonal termini lie in close apposition (see Figure 17 of the accompanying manuscript) (Aso et al., submitted). DAN input to the compartments housing the dendrites of these feedforward MBONs induces aversive and appetitive memory, respectively (this work)([98], Yamagata et al., submitted).”

*3) Related to above, in general, the role of individual lobes, the sequential use of MB lobes, the role for ongoing neural activity (prime lobes, APL, DPM) are not really incorporated into the framework provided in the discussion. Since some of the MBONs provide feedback (which is mentioned), i thought this should be better/more explicitly framed*.

Since nearly all of our behavior assays did not measure MBON function during memory consolidation, we have little new to say on this subject. Given the length of the manuscript we chose not to include extensive discussion of subjects were our data could not make a substantive contribution.

*4) Although the MBON anatomy is documented in a different study, there are many cases in this manuscript where MBON labeling in individual panels (mostly in supplement) are almost impossible to see. I think the authors should provide insets with higher magnification b/c many of the neurons are too small to see when the entire CNS is shown*.

The purpose of these illustrations was primarily to demonstrate the specificity of the expression patterns. We are proving the original confocal stacks for download at a web site referred to in the text and we added a sentence “Full confocal stacks of these images are available at www.janelia.org/split-gal4” in the Materials and Methods and the relevant Figure legends. We feel this approach will best serve the needs of expert readers.

*5) In the description of*
Figure 9
*findings, it is stated that naive responses to sugar are normal at restrictive temperature for all the lines. When I look at the figure, it appears that it is true that none are different from control, but some lines are higher and some lower than control, and my guess is that the higher lines are significantly different from the lower lines. This is a blemish in the data. Of course all papers have some blemishes, but I think the authors should state the statistical difference if there is one and just make the case that it is not a major concern*.

In the statistical analysis in Figure 9, we compared only experimental groups (GAL4/Shi) and control (+/Shi), and none of five experimental groups showed significant difference from the control. As this reviewer pointed out, 210B/shi was statistically different from 542B/shi. However, this difference does not account for memory impairment of these lines compared to the control.

6) In the sleep section, there is no rebound effect after 2 nights of sleep loss (in the lines that reduce sleep). Why?

The reviewer correctly points out that we did not consistently observe significant sleep rebound in these experiments; additional studies will be required to understand this observation. We quantified sleep rebound post dTRPA1 activation that induces sleep loss in 210B and 434B. We found sleep rebound in 210B significantly different from genotypic controls. However, this was not the case for 434B. We reason that 210B loses more sleep than 434B and hence shows more robust sleep rebound. To understand the exact quantitative relationship between amounts of sleep lost and subsequent sleep rebound, and why this may differ between wake-promoting split-GAL4 lines, will require additional studies.

*7) This study crosses many sub-fields, so citation is really not trivial to organize. But I see a few omissions where previous work was not emphasized enough (IMHO). Including these could improve the Discussion section, and will avoid inadvertently annoying a few colleagues*:

Citation omissions/mis-citations:

*In my opinion, Marin et al. Cell 2002 and Lin et al., Cell 2007 should be cited alongside Vosshall and stocker, 2007;*
[146]*;*
[136]*;*
[61]*. Another place where I see missing citations is where the authors first introduce the aversive and appetitive olfactory assays, but they only cite aversive (Tully Quinn, and some reviews). The appropriate appetitive citations should be included (Tempel original paper, and the modern versions from Waddell and Preat labs)*.

We added Marin et al. Cell 2002, Lin et al., Cell 2007 and Tempel et al., PNAS 1983. Also we added Schwaerzel et al., J Neuroscience 2003 which provides the earliest published reference for the modern version of the assay.

Previous work not emphasized enough:

*I see two corners of the literature that IMHO deserve more attention: First, the findings on glutamatergic MBONs could be discussed in the context of the published role of NMDA receptors in MB dependent learning. There are a series of studies (Saito lab, Chiang/Tully collaboration) on NMDAr that I always found hard to incorporate into a model. But now one might view these in a new light*.

We added a sentence: “In this regard, we note that previous studies demonstrated a role for NMDA receptors in olfactory memory (137, 91).”

*The other literature that I think is given short shrift is a bulk of data on the effects of internal state (satiety) on reward learning via monoaminergic modulation of MB. These papers, mostly from Waddell lab, have a major impact on how we frame and think about the current datasets and I imagine they impacted experimental design quite a bit. It might be nice to shed a bit more textual light on this*.

We added the following sentence: “Other internal states, in addition to sleep deprivation, are likely to affect the decision to carry out a particular memory-guided behavior; for example, the state of satiety has been shown to regulate memory expression (72).”

*8) Concentrations of odors for learning and for task relevant sensorimotor responses not always mentioned. Please add this info and state whether the concentrations for olfactory acuity and learning are the same*.

Missing information has been added:

“Odors were placed in cups with 30mm diameter and delivered at a flow rate of 0.6L/min per tube.”

“For all assays (training, memory test and olfactory acuity), odorants were diluted in mineral oil at a final concentration of 0.36 mM for octanol and 0.325 mM for methyl-cyclohexanol, and were delivered by a 0.4 L/min flow bubbling through odor-containing bottles.”

“Training consisted of a 10 min habituation to the training chamber with air, a 10 min presentation of odor 1 (1:36 odor:mineral oil actively streamed at 2L/min), then 10 min of odor 2 (1:36 odor:mineral oil actively streamed at 2L/min), with 60% ethanol. Air flow was matched for CS+ and CS- experiments, and ethanol was delivered by mixing pure ethanol vapor (1.5 L/min) with humidified air (1.1 L/min) at a specified ratio (135).”